# *Rpl24*[Bst] mutation suppresses colorectal cancer by promoting eEF2 phosphorylation via eEF2K

**John RP Knight**[1]*, **Nikola Vlahov**[1], **David M Gay**[1,2†], **Rachel A Ridgway**[1], **William James Faller**[1‡], **Christopher Proud**[3,4], **Giovanna R Mallucci**[5], **Tobias von der Haar**[6], **Christopher Mark Smales**[6], **Anne E Willis**[7], **Owen J Sansom**[1,2]*

[1]CRUK Beatson Institute, Garscube Estate, Glasgow, United Kingdom; [2]Institute of Cancer Sciences, University of Glasgow, Glasgow, United Kingdom; [3]Department of Biological Sciences, University of Adelaide, Adelaide, Australia; [4]Lifelong Health, South Australian Health and Medical Research Institute, Adelaide, Australia; [5]UK Dementia Research Institute at the University of Cambridge and Department of Clinical Neurosciences, University of Cambridge, Cambridge, United Kingdom; [6]School of Biosciences, Division of Natural Sciences, University of Kent, Kent, United Kingdom; [7]MRC Toxicology Unit, University of Cambridge, Cambridge, United Kingdom

*For correspondence:
j.knight@beatson.gla.ac.uk
(JRPK);
o.sansom@beatson.gla.ac.uk
(OJS)

Present address: †Biotech Research and Innovation Centre, Københavns Universitet, Københavns, Denmark; ‡NKI, Plesmanlaan, Amsterdam, Netherlands

Competing interest: The authors declare that no competing interests exist.

**Abstract** Increased protein synthesis supports the rapid cell proliferation associated with cancer. The *Rpl24*[Bst] mutant mouse reduces the expression of the ribosomal protein RPL24 and has been used to suppress translation and limit tumorigenesis in multiple mouse models of cancer. Here, we show that *Rpl24*[Bst] also suppresses tumorigenesis and proliferation in a model of colorectal cancer (CRC) with two common patient mutations, *Apc* and *Kras*. In contrast to previous reports, *Rpl24*[Bst] mutation has no effect on ribosomal subunit abundance but suppresses translation elongation through phosphorylation of eEF2, reducing protein synthesis by 40% in tumour cells. Ablating eEF2 phosphorylation in *Rpl24*[Bst] mutant mice by inactivating its kinase, eEF2K, completely restores the rates of elongation and protein synthesis. Furthermore, eEF2K activity is required for the *Rpl24*[Bst] mutant to suppress tumorigenesis. This work demonstrates that elevation of eEF2 phosphorylation is an effective means to suppress colorectal tumorigenesis with two driver mutations. This positions translation elongation as a therapeutic target in CRC, as well as in other cancers where the *Rpl24*[Bst] mutation has a tumour suppressive effect in mouse models.

## Editor's evaluation

Briefly, Knight and colleagues investigate the role of the ribosome and translational control in colorectal tumours. A mutation of a protein of the large ribosomal subunit, RPL24, is used to suppress tumours driven by two mutations found commonly in cancer, in APC and KRAS. The authors identify a mechanistic output of the RPL24 BST mutation, eEF2 phosphorylation, which they demonstrate is a major effector in inhibiting tumour cell translation and proliferation. By targeting the eEF2 kinase eEF2K, they restore protein synthesis in RPL24 mutant cells. The conclusion is well supported by the experimental data presented, which implies that translation elongation can be a potential therapeutic target of KRAS mutated CRC. Importantly, Rpl24Bst in wildtype intestine does not affect epithelial cell proliferation and differentiation, suggesting that translation elongation can be used as tumour-specific target.

## Introduction

Tumour cells require rapid protein synthesis to acquire sufficient biomass in order to divide and, as such, protein synthesis is directly regulated by many oncogenic signalling pathways (*Proud, 2019*; *Robichaud et al., 2019*; *Smith et al., 2021*). As well as exploiting protein synthesis to drive cell division, cancers use translation to selectively synthesise a proteome geared towards proliferation, survival, and immune evasion. For example, in colorectal cancer (CRC) translation of the mRNA encoding the proto-oncogene c-MYC is selectively upregulated by eIF4E and mTORC1 signalling (*Knight et al., 2020a*). Likewise, independent reports have shown that the expression of the immune suppressive ligand PD-L1 is maintained on tumour cells by the activity of the translation factors eIF4A and eIF5B as well as phosphorylation of eIF4E and eIF2α (*Suresh et al., 2020*; *Xu et al., 2019*; *Cerezo et al., 2018*).

*APC* is the most commonly mutated gene in CRC, followed by *TP53* and then *KRAS* (*Guinney et al., 2015*). We have previously shown that *Apc*-deficient mouse models of CRC are dependent on fast translation elongation, a process which can be suppressed by rapamycin leading to near complete reversal of tumorigenesis (*Faller et al., 2015*). This approach has had clinical success, where rapamycin (sirolimus) regressed *APC*-deficient polyps of familial adenomatous polyposis patients in two independent clinical trials (*Yuksekkaya et al., 2016*; *Roos et al., 2020*). Clinical data also suggest that CRCs increase translation elongation to potentiate proliferation, exemplified by the lower expression of the inhibitory kinase eEF2K correlating with worse patient survival (*Ng et al., 2019*). However, the regulation of translation elongation in CRC is complex, notably being influenced by specific cancer-associated mutations. We have shown that mutation of *Kras* drives resistance to rapamycin in *Apc*-deficient models both in terms of its effect on elongation and on proliferation (*Knight et al., 2020a*). This is consistent with *KRAS*-mutant CRCs being resistant to rapalogues, and other therapeutics (*Ng et al., 2013*; *Spindler et al., 2013*; *DeStefanis et al., 2019*) and highlights the unmet need for effective therapies against *KRAS*-mutant cancers. Indeed, a recently developed compound covalently targeting the *Kras*$^{G12C}$ mutation has shown remarkable potency against this specific mutation (*Hong et al., 2020*).

Evidence suggests that targeting translation in *KRAS*-mutant CRC can be effective. As well as our recent study re-sensitising *Kras*-mutant CRCs to rapamycin by targeting translation initiation (*Knight et al., 2020a*), we have demonstrated that *Kras*-mutant models of CRC depend upon the transporter SLC7A5 to maintain protein synthesis by facilitating the influx of amino acids (*Najumudeen et al., 2021*). These data support protein synthesis as a tractable target in CRC, with the discovery of additional factors regulating these pathways only improving the potential to target protein synthesis in the clinic (*Knight and Sansom, 2021*).

In this study, we analyse the previously characterised *Rpl24*$^{Bst}$ mutation in models of CRC with *Apc* deletion and *Kras* mutations. This spontaneously arising four nucleotide deletion in the *Rpl24* gene, which encodes RPL24 (a component of the 60S ribosomal subunit also called large ribosomal protein subunit eL24), disrupts splicing of its mRNA, effectively resulting in a *Rpl24* heterozygous animal (*Oliver et al., 2004*). Animals present with impaired dorsal pigmentation and malformed tails, among other defects, leading to the designation of a belly spot and tail (Bst) phenotype and the *Rpl24*$^{Bst}$ designation. This tool has been used to suppress overall protein synthesis in genetically engineered mouse models of c-MYC-driven B-cell lymphoma, *Pten*-deficient T-cell acute lymphoblastic leukaemia, T-cell-specific *Akt2* activation and a carcinogen-driven model of bladder cancer (*Barna et al., 2008*; *Signer et al., 2014*; *Hsieh et al., 2010*; *Jana et al., 2021*). In these studies, tumorigenesis increased total protein synthesis, which was rescued by the *Rpl24*$^{Bst}$ mutation. Suppression of protein synthesis was sufficient to slow tumorigenesis, with some *Rpl24*$^{Bst/+}$ mice surviving over three times longer than the median survival of tumour model mice wild-type for *Rpl24*. However, the means by which the *Rpl24*$^{Bst}$ mutation suppresses protein synthesis was not addressed in these studies, instead deferring to the original observation that there is likely a defect in ribosome production (*Oliver et al., 2004*).

Here, we show that decreased expression of RPL24 suppresses proliferation and extends survival in an *Apc*-deficient *Kras*-mutant pre-clinical mouse model of CRC. Importantly, we find that reduced RPL24 does not alter the available pool of ribosomal subunits, as previously suggested, but instead alters signalling that regulates a translation factor. Specifically, we observe increased phosphorylation of eEF2, an event that inhibits translation elongation. We directly measure translation elongation to show that the *Rpl24*$^{Bst}$ mutation suppresses protein synthesis at the elongation step, consistent

with increased phosphorylation of eEF2. Reducing P-eEF2 by inactivating its inhibitory kinase, eEF2K, completely restores translation elongation and protein synthesis rates as well as reversing the beneficial effect of *Rpl24^Bst* mutation in our tumour models. Interestingly, we find that the *Rpl24^Bst* mutation has no effect in *Kras* wild-type models. We attribute this to a specific requirement for physiological RPL24 in *Kras*-mutant cells, which may provide additional mechanisms to target these cells clinically.

Finally, we provide evidence from transcriptomic and proteomic analyses of patient tissue that supports the signalling pathways uncovered in our pre-clinical models being altered in the human disease. Altogether this work demonstrates that the *Rpl24^Bst* mutation is tumour suppressive in *Kras*-mutant CRC and elucidates an unexpected mode of action underlying its impact on protein synthesis. This has implications for targeting translation elongation in cancer and provides mechanistic insight to supplement the previously published efficacy of the *Rpl24^Bst* mouse in models of cancer and other diseases.

## Results

### *Rpl24^Bst* mutation does not alter intestinal homeostasis but suppresses the rate of translation

Prior to addressing the role of RPL24 in intestinal tumorigenesis we first analysed whether the *Rpl24^Bst* mutation had any effect on normal intestinal homeostasis (*Figure 1A*). We observed a reduction in RPL24 expression (*Figure 1B*), but no differences in intestinal architecture, proliferation shown by BrdU incorporation (*Figure 1B*) or abundance of stem cells (*Olfm4*), Paneth cells (lysozyme), or goblet cells (AB/PAS) (*Figure 1—figure supplement 1A*). Similarly, homeostasis in the colons of *Rpl24^Bst* mutant mice was unaffected, exemplified by no change in proliferation scored by BrdU incorporation (*Figure 1—figure supplement 1C*). In accordance with these in vivo observations, ex vivo organoids made from the small intestines of *Rpl24^Bst/+* mice established in culture and grew comparably to wild-type controls (*Figure 1C* and *Figure 1—figure supplement 2A*). Surprisingly, we measured a >40% reduction in total protein synthesis, by $^{35}$S-methionine labelling, in *Rpl24^Bst/+* organoids compared to wild-type counterparts (*Figure 1D*). Therefore, despite no change in homeostasis, *Rpl24^Bst* mutation had a dramatic effect on protein synthesis. This indicates a resistance to reduced protein synthesis in the wild-type intestine, which appears to function normally despite a dramatic reduction in protein output.

We then investigated how *Rpl24^Bst* mutation suppresses translation. Performing sucrose density gradients to quantify the number of ribosomes engaged in active translation we observed an increase in ribosomes bound to mRNAs in polysomes in *Rpl24^Bst/+* organoids, particularly the heavy polysomes (*Figure 1E*). This appears to contradict the reduction in global protein synthesis observed in *Figure 1D*. However, we and others have previously observed increased polysomes in conjunction with reduced protein synthesis in model systems where translation elongation is reduced (e.g. *Knight et al., 2015*; *Faller et al., 2015*). In these instances, slowed translation elongation increased the abundance of polysomes via changes in the phosphorylation of the elongation factor eEF2. We therefore assayed the regulatory phosphorylation of eEF2 (threonine 56/T56) in wild-type and *Rpl24^Bst* mutant samples. This phosphorylation event excludes eEF2 from the ribosome thereby impairing the translocation step of translation elongation, reducing protein synthesis (*Ryazanov and Davydova, 1989*; *Carlberg et al., 1990*).

In small intestinal tissue assayed by immunohistochemistry (IHC) we observed an increase in P-eEF2 T56, specifically in the proliferating crypt and transit amplifying zone of *Rpl24^Bst/+* mouse intestines (*Figure 1B*). Likewise, we observed a 66% increase in P-eEF2 T56 in lysates generated from *Rpl24^Bst/+* organoids compared to wild-type organoids (*Figure 1F*). These organoids also showed a 47% reduction in RPL24 expression by western blot (*Figure 1F*). Thus, in these two proliferative settings (intestinal crypts in situ and ex vivo organoids) *Rpl24^Bst* mutation increases the phosphorylation of eEF2, which is known to suppress translation. Surprisingly, we observed that in the differentiated villus, *Rpl24^Bst* mutation suppressed P-eEF2 T56 (*Figure 1B*). The reasons for this are unclear but may relate to different cell functions in the two compartments. This effect on P-eEF2 T56 is specific, as we observed no effect on the phosphorylation of other translation related proteins, 4E-BP1 or RPS6 (at S240/S244), readouts for modulation of signalling downstream of mTORC1 (*Figure 1—figure supplement 1A, B*).

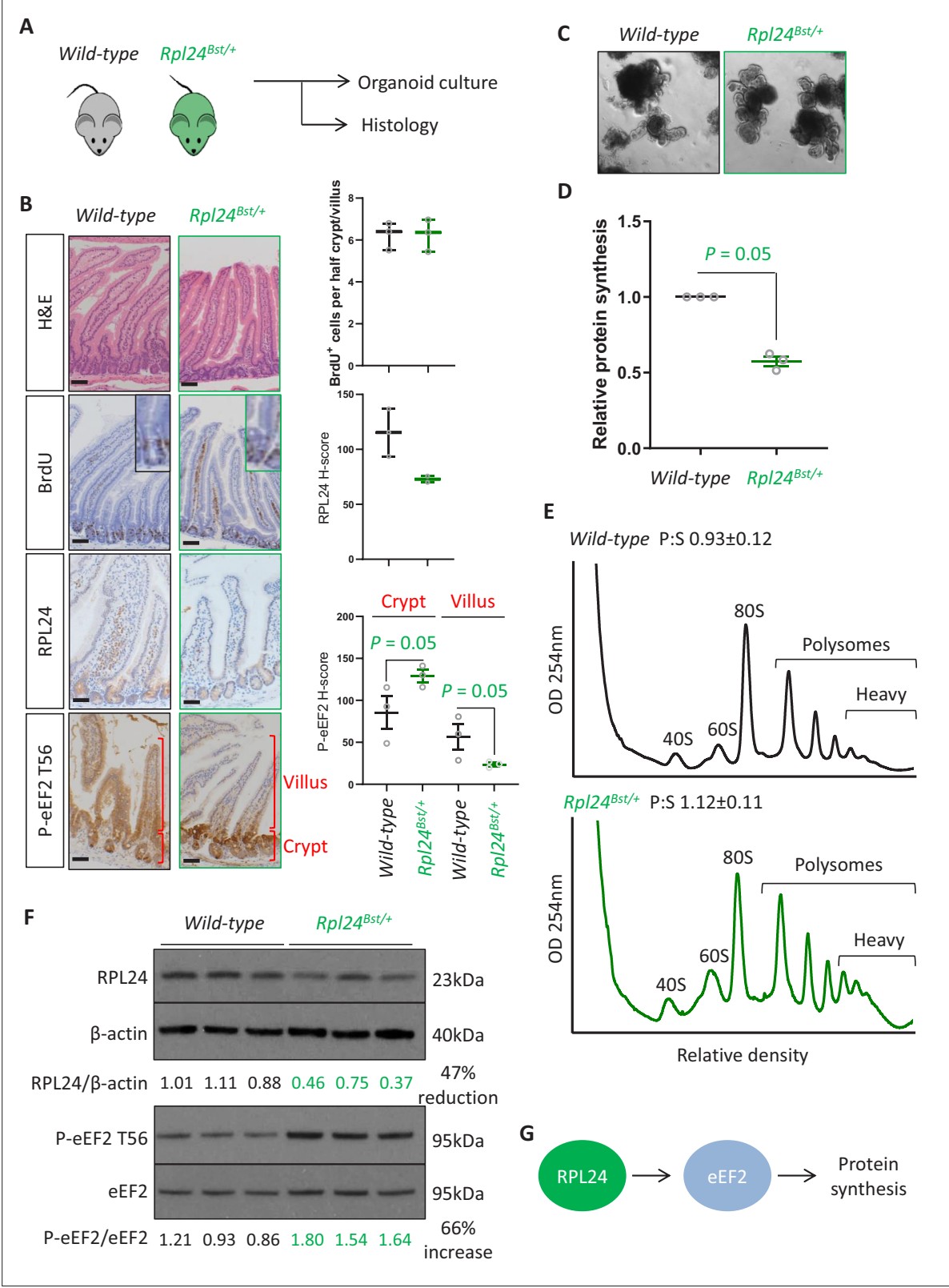

**Figure 1.** *Rpl24^Bst* mutation slows translation elongation but does not affect homeostasis in the intestinal epithelium. (**A**) Schematic representation of experimental procedure. Intestines from wild-type or *Rpl24^Bst/+* mice were analysed by histology or processed to make intestinal organoids. (**B**) Staining for H&E, BrdU, RPL24, and P-eEF2 T56 in sections from the small intestines of wild-type and *Rpl24^Bst/+* mice. Red brackets in P-eEF2 staining indicate crypts and villi, corresponding to quantification to the right. Bars represent 50 μm. Graphs on the right show scoring for BrdU-positive cells, and *H-*

*Figure 1 continued on next page*

*Figure 1 continued*

score calculated for RPL24 and P-eEF2 T-56, plotted ± standard error of the mean (SEM). Significance was determined by one-tailed Mann–Whitney *U* test. (**C**) Micrographs of small intestinal organoids generated from wild-type or *Rpl24^Bst/+^* mice. (**D**) Protein synthesis rate quantified by ³⁵S-methionine incorporation in wild-type or *Rpl24^Bst/+^* organoids (*n* = 3), expressed relative to the wild-type protein synthesis rate (=1). Data are from three biologically independent organoid lines for each genotype represented ± SEM with significance determine by Mann–Whitney *U* test. (**E**) Representative polysome profiling from wild-type or *Rpl24^Bst/+^* organoids. Average polysome:subpolysome ratios from three independent organoid lines per genotype are shown above each profile ± SEM. (**F**) Western blotting from protein lysates generated from three biologically independent organoid lines for each genotype. Values for RPL24 expression relative to β-actin and P-eEF2 T56 relative to eEF2 are shown under each lane. The average of the wild-type lanes in both cases has been set to 1. There is a 47% reduction in RPL24 and a 66% increase in P-eEF2 T56 in the *Rpl24^Bst/+^* organoids. (**G**) Schematic of the potential role of RPL24 in regulating protein synthesis via eEF2. All scale bars are 50 µm.

The online version of this article includes the following figure supplement(s) for figure 1:

**Source data 1.** Top: data from *Figure 4F*.

**Figure supplement 1.** *Rpl24^Bst^* mutation does not affect intestinal homeostasis.

**Figure supplement 2.** *Rpl24^Bst^* mutation does not alter proliferation in organoids or the 60S:40S ratio, but does suppress regeneration post irradiation.

Similarly, we saw no change in the phosphorylation of the translation stress marker eIF2α in *Rpl24^Bst^* mutant mice (*Figure 1—figure supplement 1A, B*).

Altogether, this analysis of wild-type tissue indicates that physiological RPL24 expression is not required for proliferation or function, but reduction of RPL24 expression reduces the rate of translation. Interestingly, this involves regulation of translation elongation and correlates with increased P-eEF2 (*Figure 1G*). Previously the *Rpl24^Bst^* mutation had been suggested to suppress ribosome biogenesis, resulting in uneven 40S and 60S ratios. However, we observed no alteration in the relative levels of the 40S and 60S subunits in sucrose density gradients (*Figure 1—figure supplement 2B*), consistent with previous reports that RPL24 deletion has negligible effect on ribosome biogenesis (*Barkić et al., 2009*).

The wild-type mouse intestine regenerates following γ-irradiation, dependent on Wnt and mitogen-activated protein kinase (MAPK) signalling pathways. These pathways are often deregulated in colorectal tumours, such that this intestinal regeneration acts as a surrogate for oncogenic potential, with reduced regenerative capacity indicative of reduced tumorigenic proliferation (*Faller et al., 2015*). We observe that mutation of *Rpl24* restricts regeneration of the small intestine (*Figure 1—figure supplement 2C*). Thus, RPL24 expression enables regeneration, which may correlate with effects in tumorigenesis.

## RPL24 is required for proliferation in *Apc*-deficient *Kras*-mutant intestinal tumours

Next, we analysed the effect of the *Rpl24^Bst^* mutation on a model of CRC driven by tamoxifen inducible *Villin^CreER^*-mediated deletion of *Apc* and activation of KRAS with a G12D mutation. This model can be used with homozygous deletion of *Apc* (*Apc^fl/fl^*) where intestinal hyperproliferation generates a short-term (3–4 days) model or with heterozygous deletion of *Apc* (*Apc^fl/+^*) where intestinal adenomas form following spontaneous loss of the second copy of *Apc*. The recombination of a lox-STOP-lox allele at the endogenous *Kras* locus expresses a constitutively active G12D mutant form of the protein. Mice were generated with the *Apc^fl/fl^* and *Kras^G12D^* alleles with and without the *Rpl24^Bst^* mutation (*Figure 2A*). Hyperproliferation in the *Apc^fl/fl^ Kras^G12D/+^ Rpl24^Bst/+^* mutant mice was significantly suppressed compared to *Apc^fl/fl^ Kras^G12D/+^* control mice (*Figure 2B, C*). Reduced RPL24 expression was confirmed by IHC (*Figure 2C*) and coincided with increased P-eEF2 throughout the proliferative crypt area of the *Apc^fl/fl^ Kras^G12D/+^ Rpl24^Bst/+^* intestine (*Figure 2C* and *Figure 2—figure supplement 1*). Hyperproliferation in the colon mirrored that of the small intestine, with reduced proliferation in mice mutant for *Rpl24* compared to those expressing wild-type levels of RPL24 (*Figure 2—figure supplement 1*). In parallel, organoids were derived from the small intestines of the same genotypes (*Apc^fl/fl^ Kras^G12D/+^* and *Apc^fl/fl^ Kras^G12D/+^ Rpl24^Bst/+^*) and their growth compared. *Rpl24^Bst^* mutation resulted in significantly less proliferation ex vivo (*Figure 2D*), consistent with the in vivo experiment shown in *Figure 2B*.

In the tumour model, where a single copy of *Apc* is deleted, *Apc^fl/+^ Kras^G12D/+^ Rpl24^Bst/+^* mice lived on average 32 days longer than *Apc^fl/+^ Kras^G12D/+^* controls, an extension of survival of 45% (*Figure 2E*). Furthermore, *Apc^fl/+^ Kras^G12D/+^ Rpl24^Bst/+^* organoids derived from the adenomas in this tumour model

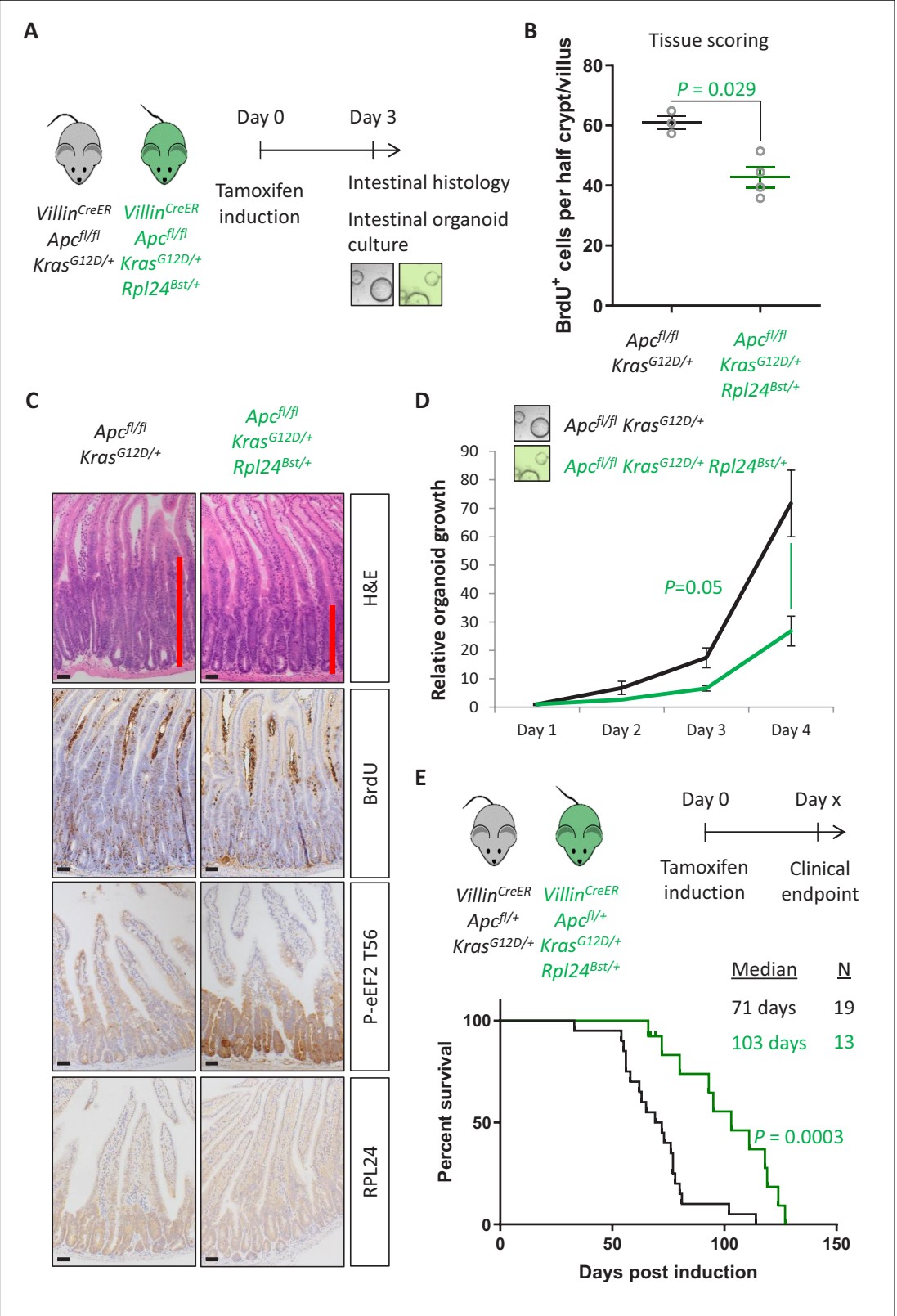

**Figure 2.** *Rpl24*[Bst] mutation suppresses proliferation and extends survival in an *Apc*-deficient *Kras*-mutant mouse model of colorectal cancer (CRC). (**A**) Schematic representation of experimental protocols. *Villin*[CreER] *Apc*[fl/fl] *Kras*[G12D/+] or *Villin*[CreER] *Apc*[fl/fl] *Kras*[G12D/+] *Rpl24*[Bst/+] mice were induced by intraperitoneal injection of tamoxifen at 80 mg/kg then intestinal tissue analysed 3 days later. Tissue was taken for histological analysis or processed into intestinal organoids. (**B**) Quantification of BrdU incorporation in small intestinal crypt/villus axes following deletion of *Apc* and activation of *Kras*,

*Figure 2 continued on next page*

*Figure 2 continued*

with (*n* = 4) or without (*n* = 3) *Rpl24^Bst^* mutation. Data are represented as the mean number of BrdU-positive cells per half crypt/villus from >20 axes per mouse ± standard error of the mean (SEM). Significance was determined by Mann–Whitney *U* test. (**C**) Representative images of intestines from the same experiment as in (**B**), stained for H&E, BrdU, P-eEF2 T56, and RPL24. The red bar on the H&E images indicates the extent of the proliferative zone. Bars represent 50 µm. (**D**) *Apc^fl/fl^ Kras^G12D/+^* organoids with or without *Rpl24^Bst^* mutation were grown for 4 days and growth relative to day 1 determined by Cell-Titer Blue assay. Data show the mean ± SEM of *n* = 3 independent organoid lines. Significance was determined by one-tailed Mann–Whitney *U* test. (**E**) Top: schematic of experimental protocol. *Villin^CreER^ Apc^fl/+^ Kras^G12D/+^* or *Villin^CreER^ Apc^fl/+^ Kras^G12D/+^ Rpl24^Bst/+^* mice induced with 80 mg/kg tamoxifen then monitored until clinical endpoint. Survival plot for these genotypes for the days post-induction that they reached endpoint. The median survival and *n* number for each cohort are shown and significance determined by Mantel–Cox test. Censored subjects were removed from the study due to non-intestinal phenotypes. All scale bars are 50 µm.

The online version of this article includes the following figure supplement(s) for figure 2:

**Figure supplement 1.** *Rpl24Bst* mutation leads to increased eEF2 phosphorylation.

**Figure supplement 2.** *Rpl24^Bst^* mutation suppresses proliferation in cells from colorectal cancer models.

grew more slowly than controls (*Figure 2—figure supplement 2*). There was no significant difference in the number or volume of tumours at experimental endpoint (*Figure 2—figure supplement 2*), indicating that adenomas can form but take longer to reach a clinically significant burden. Therefore, RPL24 enables proliferation in *Apc*-deficient, KRAS-activated cells within the intestinal epithelium of the mouse.

## RPL24 maintains translation elongation in *Apc*-deficient *Kras*-mutant intestinal tumour models

The suppression of tumorigenesis in the *Apc*-deficient *Kras*-mutant model correlated with increased phosphorylation of eEF2 (*Figure 2C* and *Figure 2—figure supplement 1*). Increased P-eEF2 was not accompanied by increased P-eIF2α, which controls translation initiation in response to various stress signals (*Figure 2—figure supplement 1*). This indicates that stress signalling to eIF2α was not influenced by the *Rpl24^Bst^* mutation, highlighting the specificity in the RPL24-dependent regulation of eEF2. To investigate this further we used three methods to measure the rate of translation: polysome profiling, ^35^S-methionine labelling, and harringtonine run-off assays (*Figure 3A*). Polysome profiling from extracted crypts from *Apc^fl/fl^ Kras^G12D/+^* and *Apc^fl/fl^ Kras^G12D/+^ Rpl24^Bst/+^* mice showed an increase in polysomes with the *Rpl24* mutation (*Figure 3B*), and notably a significant increase in the quantity of heavy polysomes (*Figure 3C*). Intestinal organoids of the same genotype showed a 35% reduction in ^35^S-methionine incorporation (*Figure 3D*). These same organoids had a greater than 40% decrease in elongation rate measured by harringtonine run-off (*Figure 3E* and *Figure 3—figure supplement 1A*). Together these data provide compelling evidence that normal RPL24 expression is required to maintain translation elongation in this CRC model. Polysome profiles and protein synthesis rate measurements from *Apc^fl/+^ Kras^G12D/+^ Rpl24^Bst/+^* adenoma cultures also showed more polysomes and lower protein synthesis compared to control adenoma cultures (*Figure 3—figure supplement 1B, C*). The reduced protein synthesis rate does not correlate with differences in free ribosomal subunit availability as the ratio of 40S to 60S subunits is unchanged by the *Rpl24* mutation in *Apc*-deficient *Kras*-mutant cells (*Figure 3—figure supplement 1D*). To investigate this further we assayed selected ribosomal protein abundances in *Apc^fl/fl^ Kras^G12D/+^* and *Apc^fl/fl^ Kras^G12D/+^ Rpl24^Bst/+^* organoids by western blot. RPL10 expression was increased by the *Rpl24* mutation, RPL22 was decreased, while RPS6 levels were unchanged (*Figure 3—figure supplement 1E*). While the significance of these individual changes is unknown, the data show that there is not a global suppression of ribosomal protein expression in *Rpl24* mutant cells, consistent with no change in free ribosomal subunit levels.

The reduced expression of RPL24, but maintained ribosomal subunit stoichiometry, in *Rpl24^Bst/+^* mice raises the possibility of heterogeneous ribosomes (reviewed by *Gay et al., 2021*), with some potentially lacking RPL24. To address this, we purified protein from sucrose density gradients from *Apc^fl/fl^ Kras^G12D/+^* and *Apc^fl/fl^ Kras^G12D/+^ Rpl24^Bst/+^* organoids and analysed ribosomal protein expression within subpolysomes and polysomes (*Figure 3—figure supplement 2*). The large subunit protein RPL10 showed little change in distribution between the two genotypes, consistent with no change in free ribosomal subunit levels and a shift from light to heavy polysomes, but similar overall polysome number. Importantly, RPL24 can incorporate into 60S, 80S, and polysomes in *Apc^fl/fl^ Kras^G12D/+^*

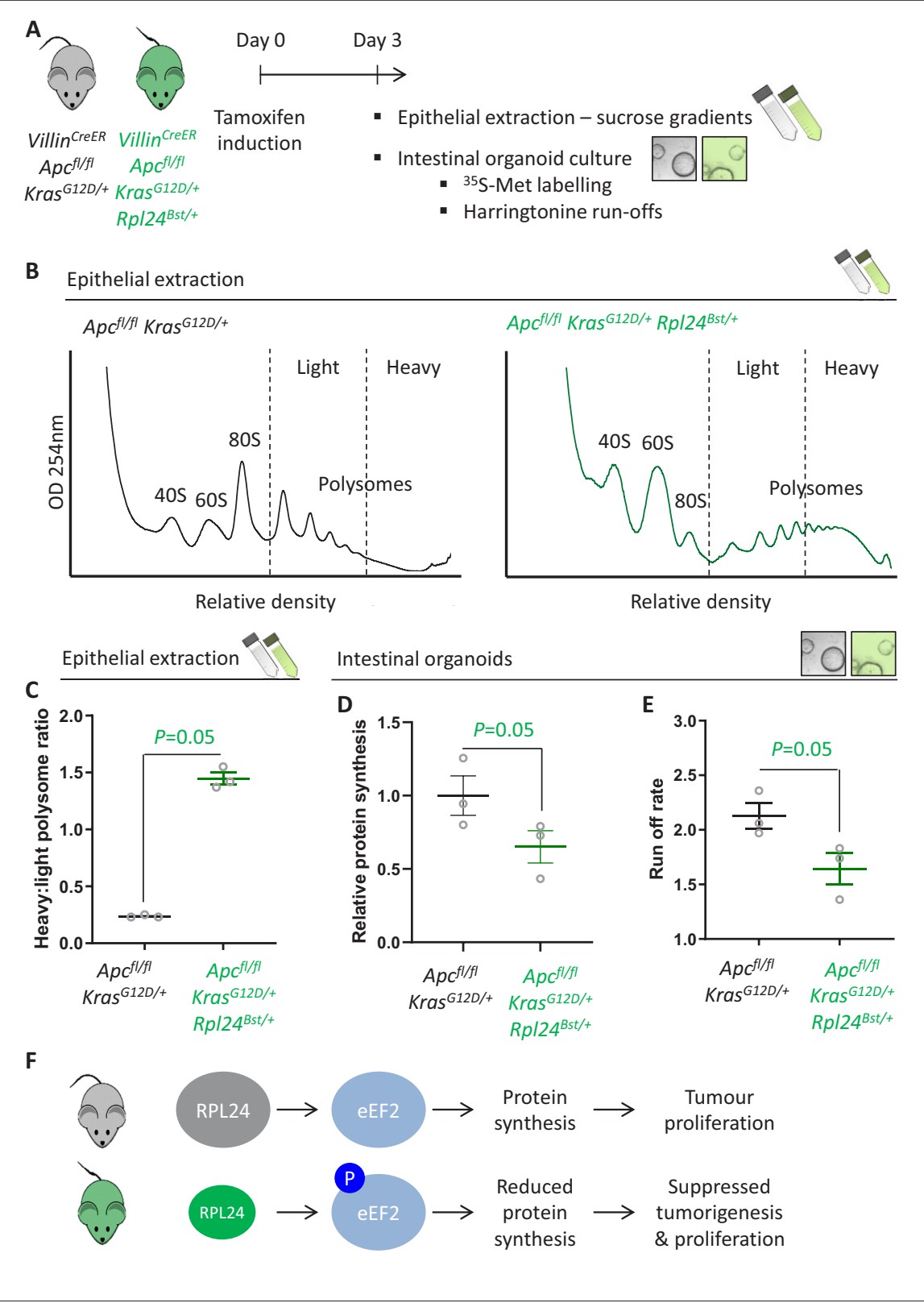

**Figure 3.** *Rpl24*[Bst] mutation slows translation elongation in *Apc*-deficient *Kras*-mutant mouse models of colorectal cancer (CRC). (**A**) Schematic representation of experimental approach. *Villin*[CreER] *Apc*[fl/fl] *Kras*[G12D/+] or *Villin*[CreER] *Apc*[fl/fl] *Kras*[G12D/+] *Rpl24*[Bst/+] mice were induced by intraperitoneal injection of tamoxifen at 80 mg/kg then intestinal tissue analysed 3 days later. Intestines were enriched for crypt epithelium for sucrose density analysis or processed into intestinal organoids. (**B**) Representative sucrose density polysome profiles generated from *Apc*[fl/fl] *Kras*[G12D/+] intestinal extracts with or

*Figure 3 continued on next page*

*Figure 3 continued*

without the *Rpl24*<sup>Bst</sup> mutation. Subpolysomal components (40S, 60S, and 80S) and polysomes are labelled, with the polysomes also split pictorially into light and heavy. (C) Quantification of the heavy:light polysome ratio from the experiment in (B). Data show the mean of analysis from three mice ± standard error of the mean (standard error of the mean, SEM) with significance determined by one-tailed Mann–Whitney *U* test. (D) Relative protein synthesis rate quantified by $^{35}$S-methionine incorporation in *Apc*<sup>fl/fl</sup> *Kras*<sup>G12D/+</sup> three biologically independent organoid lines either wild-type or mutant for *Rpl24*<sup>Bst</sup>. Data are represented ± SEM with significance determine by Mann–Whitney *U* test. (E) Ribosome run-off rate determined in *Apc*<sup>fl/fl</sup> *Kras*<sup>G12D/+</sup> small intestinal organoid lines either wild-type or mutant for *Rpl24*<sup>Bst</sup> (n = 3 per genotype). Data are represented as the mean of three biological replicates ± SEM with significance determine by Mann–Whitney *U* test. Raw data are available in *Figure 3—figure supplement 1A*. (F) Schematics of the regulation of protein synthesis and tumour proliferation downstream of RPL24. Smaller RPL24 in bottom scheme represents reduced RPL24 expression. 'P' represents phosphorylation of eEF2.

The online version of this article includes the following source data and figure supplement(s) for figure 3:

**Figure supplement 1.** The effect of *Rpl24*<sup>Bst</sup> mutation on translation and ribosome composition.

**Figure supplement 1—source data 1.** Left: data from *Figure 3—figure supplement 1E*.

**Figure supplement 2.** Assocation of ribosomal proteins with the ribosomes in *Rpl24*<sup>Bst</sup> mutant and wild-type organoids.

**Figure supplement 2—source data 1.** Top: data from *Figure 3—figure supplement 2*.

*Rpl24*<sup>Bst/+</sup> organoids. The distribution of RPL24 appeared altered, with less in the 60S and more in the polysomes, although this was not significant. It is difficult to interpret absolute amounts of protein in each fraction and to compare this between genotypes. However, considering that *Apc*<sup>fl/fl</sup> *Kras*<sup>G12D/+</sup> *Rpl24*<sup>Bst/+</sup> organoids express less RPL24 (*Figure 3—figure supplement 1E*) the reduction in RPL24 in 60S subunits is even more striking. Thus, our data do not rule out the possibility of ribosome hetero-geneity in that some 60S subunits in *Apc*<sup>fl/fl</sup> *Kras*<sup>G12D/+</sup> *Rpl24*<sup>Bst/+</sup> organoids may lack RPL24.

Interpreting this molecular analysis in conjunction with the effects on tumorigenesis leads to the conclusion that RPL24 expression maintains translation elongation and protein synthesis rates, which in turn maintain tumour-related proliferation (*Figure 3F*). Suppressing RPL24 expression increases P-eEF2, an effect also seen in lysates from *Apc*<sup>fl/fl</sup> *Kras*<sup>G12D/+</sup> *Rpl24*<sup>Bst/+</sup> organoids (*Figure 3—figure supplement 1E*), which decreases the rate of elongation and overall protein synthesis and correlates with suppressed tumorigenesis and proliferation in vivo.

## *Rpl24* mutation has no effect in CRC models expressing wild-type *Kras*

In parallel to analysing the effect of *Rpl24* mutation in *Apc*-deficient *Kras*-mutant intestinal tumours, we also assessed its role in *Apc*-deficient models wild-type for *Kras*. We have previously shown that these are dependent on signalling from mTORC1 to maintain low levels of P-eEF2, and that rapamycin induces P-eEF2 to great therapeutic benefit (*Faller et al., 2015*). We observed that the hyperprolifera-tion in the small intestine or colon of *Apc*<sup>fl/fl</sup> mice was not reduced by the *Rpl24*<sup>Bst</sup> mutation (*Figure 4A, B* and *Figure 4—figure supplement 1A*). Furthermore, in both germline *Apc*<sup>Min/+</sup> and inducible *Apc*<sup>fl/+</sup> models of *Apc* deficiency we see no benefit of the *Rpl24*<sup>Bst</sup> mutation, with no difference in survival or tumour development (*Figure 4C* and *Figure 4—figure supplement 1B*). In agreement, *Apc*<sup>fl/fl</sup> organoids with the *Rpl24*<sup>Bst</sup> mutation grew at an identical rate to those wild-type for *Rpl24* in culture (*Figure 4D*). Together these data demonstrate that reduced RPL24 expression does not limit tumori-genesis in *Apc*-deficient CRC models with wild-type *Kras*.

Despite no effect on proliferation, we observed an increase in P-eEF2 in *Apc*<sup>fl/fl</sup> *Rpl24*<sup>Bst/+</sup> intestines compared to *Apc*<sup>fl/fl</sup> (*Figure 4B*), showing that P-eEF2 is consistently increased in the intestinal crypts of *Rpl24*<sup>Bst/+</sup> mice. However, we detected no change in the ratio of polysomes to subpolysomes in *Apc*<sup>fl/fl</sup> *Rpl24*<sup>Bst/+</sup> intestines compared to *Apc*<sup>fl/fl</sup> (*Figure 4—figure supplement 2A, B*) and no change in protein synthesis rate between organoids of these same genotypes (*Figure 4E*). In contrast, rapamycin treatment significantly reduces protein synthesis in *Apc*<sup>fl/fl</sup> organoids treated in parallel (*Figure 4E*). From these data, we conclude that the change in P-eEF2 does not limit the rate of protein synthesis which allows efficient tumorigenesis in these *Kras* wild-type models of CRC. There was no alteration in the phosphorylation status of 4E-BP1, RPS6, or eIF2α in the *Rpl24*<sup>Bst</sup> mutants in the *Apc*<sup>fl/fl</sup> model (*Figure 4—figure supplement 2C*), indicating that translation promoting mTORC1 signalling remains high while translation stress signalling to eIF2α is unchanged.

We hypothesised that the reason for the KRAS specificity seen with the *Rpl24*<sup>Bst</sup> mutation may relate to expression levels between the different genotypes analysed. Using unbiased RNA sequencing data

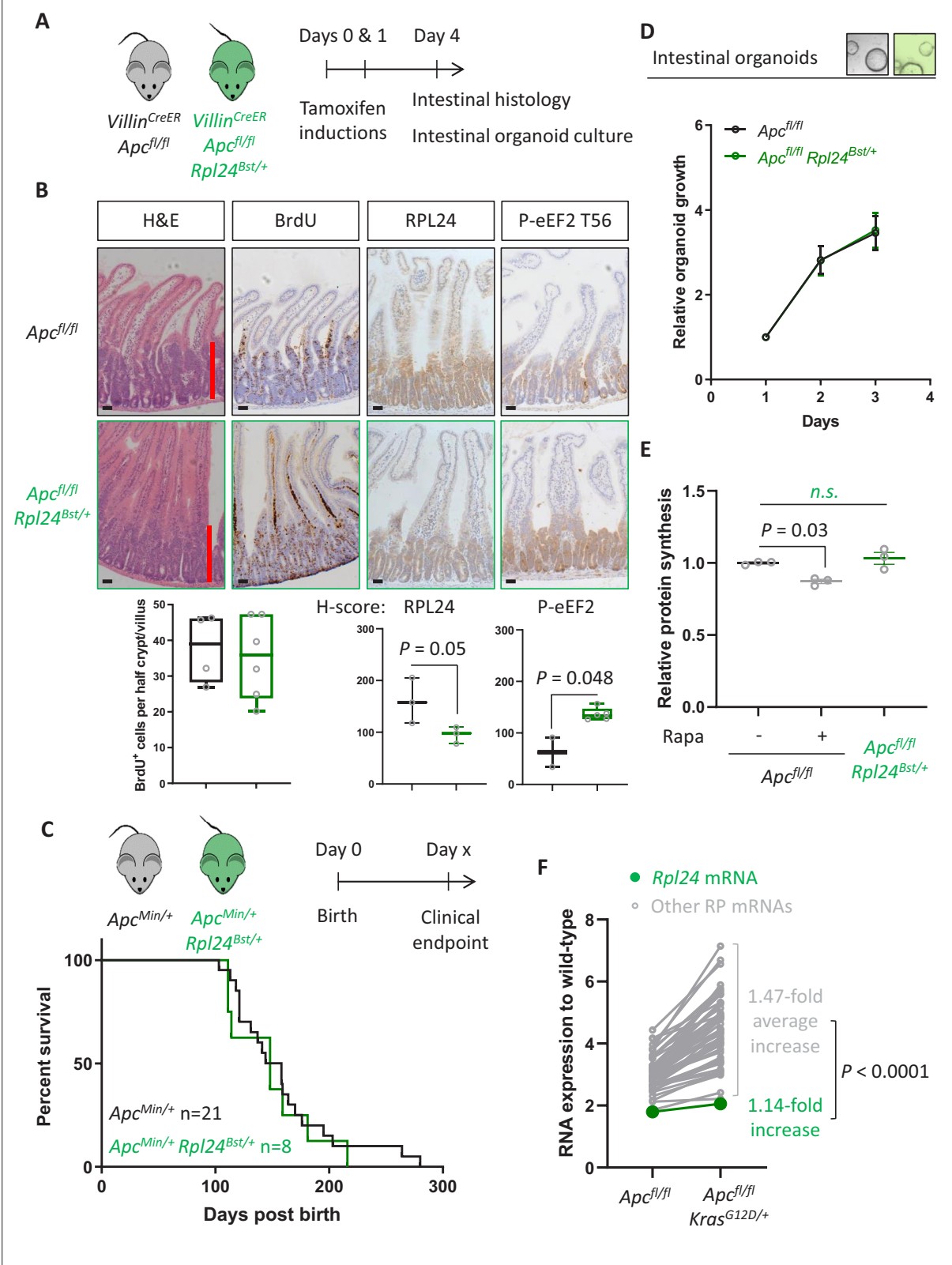

**Figure 4.** *Rpl24$^{Bst}$* mutation does not suppress proteins synthesis or proliferation in *Apc*-deficient *Kras* wild-type mouse models of colorectal cancer (CRC). (**A**) Schematic representation of experimental approach. *Villin$^{CreER}$ Apc$^{fl/fl}$* or *Villin$^{CreER}$ Apc$^{fl/fl}$ Rpl24$^{Bst/+}$* mice were induced by two intraperitoneal injection of tamoxifen at 80 mg/kg on days 0 and 1 then intestinal tissue analysed on day 4. Intestines were analysed histologically or intestinal organoids generated. (**B**) Top: representative micrographs showing proliferation as BrdU positivity and extent of proliferation as a red bar in H&E

*Figure 4 continued on next page*

*Figure 4 continued*

image. RPL24 and P-eEF2 T56, staining is also shown for each genotype. Bars represent 50 μm. Below: BrdU scoring from *Apc^fl/fl* or *Apc^fl/fl Rpl24^Bst/+* mouse intestines and *H*-scores for RPL24 and P-eEF2 T56 protein levels. For BrdU scoring BrdU was administered 2 hr before sampling and at least 20 half crypt/villus axes were scored per animal and the mean plotted ± standard error of the mean (standard error of the mean, SEM). (**C**) *Apc^Min/+* tumour model survival curve, for mice with and without *Rpl24^Bst* mutation. Lack of a significant difference was determined by Mantel–Cox test. (**D**) Relative growth of *Apc^fl/fl* and *Apc^fl/fl Rpl24^Bst/+* small intestinal organoids over 3 days, measure by Cell-Titer Blue assay. The average change in proliferation is plotted from three independent biological replicates per genotype. (**E**) Relative protein synthesis rates quantified from ^35S-methionine incorporation into *Apc^fl/fl*, *Apc^fl/fl* treated with 250 nM rapamycin for 24 hr and *Apc^fl/fl Rpl24^Bst/+* small intestinal organoids. Significant changes were calculated by one-way analysis of variance (ANOVA) with Tukey's multiple comparison. *N* = 3 per genotype with the mean protein synthesis rate for each genotype plotted ± SEM. (**F**) Relative expression of ribosomal protein mRNAs in *Villin^CreER Apc^fl/fl* and *Villin^CreER Apc^fl/fl Kras^G12D/+* whole intestine samples, where wild-type tissue has been normalised to 1. The fold increase in expression from *Apc^fl/fl* to *Apc^fl/fl Kras^G12D/+* samples for *Rpl24* and the average of all other RP mRNAs are shown. Statistical analysis was by one sample *t*-test of the other RP mRNA fold changes using the fold change for *Rpl24* mRNA as the hypothetical mean. All scale bars are 50 μm.

The online version of this article includes the following figure supplement(s) for figure 4:

**Source data 1.** Data relate to *Figure 4F* and *Figure 4—figure supplement 2D*.

**Figure supplement 1.** *Rpl24^Bst* mutation has no benefit in a model of colorectal cancer (CRC) with wild-type *Kras*.

**Figure supplement 2.** *Rpl24^Bst* mutation has no effect on polysomes or some signaling pathways.

---

from wild-type, *Apc^fl/fl* and *Apc^fl/fl Kras^G12D/+* small intestinal tissue we observed a consistent increase in ribosomal protein expression following *Apc* deletion, then again following KRAS activation (*Figure 4—figure supplement 2D*). This is consistent with previous reports (*Smit et al., 2020*), and a requirement to increase protein synthesis as a direct consequence of KRAS activation. Indeed, the mRNAs for all ribosomal proteins with sufficient reads were increased on average nearly 1.5-fold by KRAS activation in the small intestine (*Figure 4F*). In contrast, the *Rpl24* mRNA was only increased by 1.14-fold following Kras mutation, despite a nearly twofold increase following deletion of *Apc* (*Figure 4F*). This manifests as a significant difference in the RNA expression of *Rpl24* compared to the other ribosomal proteins. Therefore, RPL24 expression may be sufficient in *Rpl24^Bst* mice in *Apc*-deleted models, but then becomes limiting following *Kras* mutation due to the limited upregulation of *Rpl24* expression accompanying KRAS activation.

## Genetic inactivation of eEF2K completely reverses the anti-proliferative benefit of *Rpl24* mutation

Thus far we have demonstrated a correlation between the increase in P-eEF2 and the slowing of translation elongation following *Rpl24^Bst* mutation. To test whether the slowing of elongation caused by *Rpl24^Bst* mutation was dependent on P-eEF2, we used a whole-body point mutant of eEF2K, the kinase that phosphorylates eEF2, which almost completely inactivates its kinase activity (*Gildish et al., 2012*). We crossed this *Eef2k^D273A/D273A* allele to the *Apc^fl/fl Kras^G12D/+ Rpl24^Bst/+* mice to generate *Apc^fl/fl Kras^G12D/+ Rpl24^Bst/+ Eef2k^D273A/D273A* mice. In the short-term hyperproliferation model, the inactivation of *Eef2k* completely reversed the suppression of proliferation seen in *Apc^fl/fl Kras^G12D/+ Rpl24^Bst/+* small intestines (*Figure 5A–C*), and the medial colon (*Figure 5—figure supplement 1A*). The kinase inactive *Eef2k* allele resulted in P-eEF2 being undetectable (*Figure 5C* and *Figure 2—figure supplement 1*), and we previously reported no difference in hyperproliferation between *Apc^fl/fl Kras^G12D/+* and *Apc^fl/fl Kras^G12D/+ Eef2k^D273A/D273A* models (*Knight et al., 2020a*). The reversal in proliferation rate with *Eef2k* and *Rpl24^Bst* mutations was also seen in intestinal organoid growth after 3 days (*Figure 5D*). Furthermore, inactivation of eEF2K reverted the survival benefit of the *Rpl24^Bst* mutation in the *Apc^fl/+ Kras^G12D/+* tumour model (*Figure 5E*). This experiment also shows the lack of effect of the *Eef2k^D273A/D273A* mutation on tumorigenesis. Indeed, *Eef2k^D273A/D273A* had no impact on tumorigenesis in inducible *Apc^fl/+* and germline *Apc^Min/+* models of *Apc*-deficient intestinal tumorigenesis (i.e., expressing wild-type *Kras*) (*Figure 5—figure supplement 1B, C*). Consistent with this, the *Eef2k^D273A/D273A* allele had no effect on proliferation in the *Apc^fl/fl* hyperproliferation model, either alone or in combination with *Rpl24^Bst* mutation in both the small intestine and colon (*Figure 5—figure supplement 1D*). Heterozygous inactivation of *Eef2k* resulted in a slight reversal of the extension of survival associated with *Rpl24^Bst* mutation in the *Apc^fl/+ Kras^G12D/+* tumour model (*Figure 5—figure supplement 2A*).

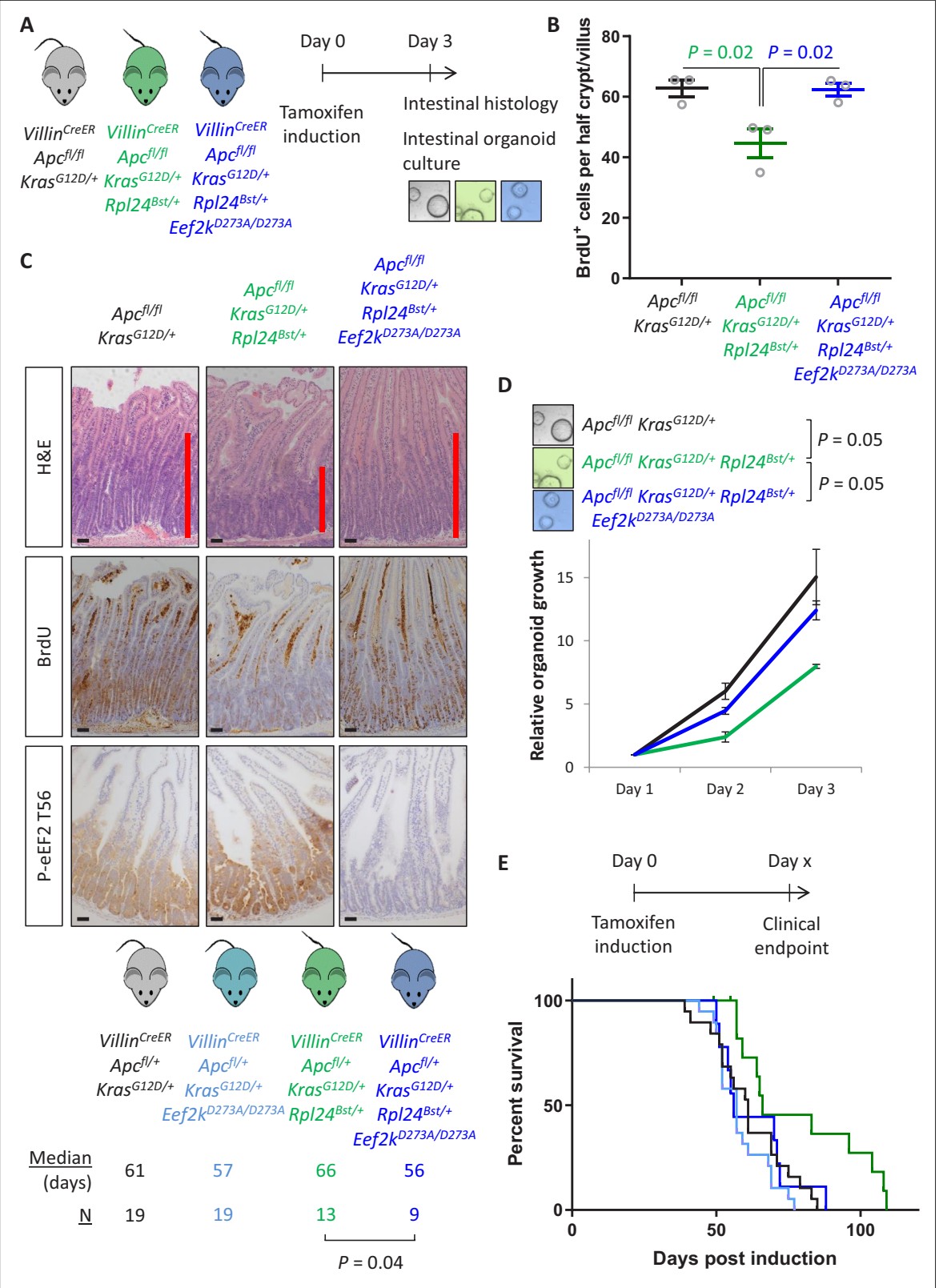

**Figure 5.** Genetic inactivation of *Eef2k* reverses the reduced tumorigenesis following *Rpl24^Bst* mutation in *Apc*-deficient *Kras*-mutant models of colorectal cancer (CRC). (**A**) Schematic representation of experimental approach. *Villin^CreER Apc^fl/fl Kras^G12D/+*, *Villin^CreER Apc^fl/fl Kras^G12D/+ Rpl24^Bst/+*, or *Villin^CreER Apc^fl/fl Kras^G12D/+ Rpl24^Bst/+ Eef2k^D273A/D273A* mice were induced by intraperitoneal injection of tamoxifen at 80 mg/kg then intestinal tissue analysed 3 days later. Intestines were analysed histologically or processed into intestinal organoids. (**B**) BrdU incorporation quantified from within small

*Figure 5 continued on next page*

**Figure 5 continued**

intestinal crypt/villus axes following deletion of *Apc* and activation of *Kras*, either wild-type of mutant for *Rpl24*, or mutant for *Rpl24* and *Eef2k*. Data are represented as the mean of at least 20 crypt/villi per mouse ± standard error of the mean (SEM) with significance determined by one-way analysis of variance (ANOVA) with Tukey's multiple comparison. *N* = 3 per genotype. (**C**) Representative images of H&E, BrdU, and P-eEF2 T56 staining of intestines from the same experiment as (**B**). Red bar on H&E indicates extent of proliferative zone. Bars represent 50 µm. (**D**) Organoids deficient for *Apc* and with activated *Kras* with or without *Rpl24^Bst^* mutation, or mutant for both *Rpl24* and *Eef2k* were grown for 3 days and growth relative to day 1 determined by Cell-Titer Blue assay. Data show the mean ± SEM of *n* = 3 biologically independent organoid lines. Significance was determined by one-tailed Mann–Whitney *U* test. (**E**) Survival plot for *Apc Kras* ageing mice with or without the *Rpl24^Bst^* mutation, *Eef2K* mutation and with both *Rpl24* and *Eef2k* mutations. Median survival and *n* numbers for each cohort are shown and significance determined by Mantel–Cox test. Censored subjects were removed from the study due to non-intestinal phenotypes. All scale bars are 50 µm.

The online version of this article includes the following figure supplement(s) for figure 5:

**Figure supplement 1.** *Eef2k^D273A/D273A^* mutation has no effect on tumorigenesis in *Kras* wild-type models.

**Figure supplement 2.** Heterozygous mutation of *Eef2k^D273A/+^* partially suppresses the effects of *Rpl24^Bst^* mutation in the tumour model.

Therefore, mutation of *Rpl24* requires functional eEF2K to suppress proliferation and extend survival in this model of CRC. There was no alteration in the number or cumulative size of tumours in the ageing model at endpoint (**Figure 5—figure supplement 2B**), again identifying tumour cell proliferation, rather than tumour initiation, as the principal factor regulated by RPL24 and eEF2K. These data also confirm that the effect of *Rpl24^Bst^* mutation on the tumour phenotype is entirely dependent upon eEF2K activity.

### *Rpl24*^Bst^ mutation suppresses translation exclusively via eEF2K/P-eEF2

Next, we addressed the molecular consequences of inactivation of eEF2K downstream of *Rpl24^Bst^* mutation. The reduction in protein synthesis that results from *Rpl24^Bst^* mutation is completely reversed by inactivating eEF2K (**Figure 6A**), with *Apc^fl/fl^ Kras^G12D/+^ Rpl24^Bst/+^ Eef2k^D273A/D273A^* organoids having an almost identical translation capacity as *Apc^fl/fl^ Kras^G12D/+^* controls. Similarly, the rate of ribosome run-off was also reverted in *Apc^fl/fl^ Kras^G12D/+^ Rpl24^Bst/+^ Eef2k^D273A/D273A^* compared to *Apc^fl/fl^ Kras^G12D/+^ Rpl24^Bst/+^* organoids, again to the same rate as controls with wild-type *Rpl24* (**Figure 6B** and **Figure 6—figure supplement 1A**). In agreement, crypt fractions from *Apc^fl/fl^ Kras^G12D/+^ Rpl24^Bst/+^ Eef2k^D273A/D273A^* mice have a reduced, although not significant, the number of heavy polysomes compared to *Apc^fl/fl^ Kras^G12D/+^ Rpl24^Bst/+^* crypt cells (**Figure 6—figure supplement 2B**), indicating faster translation elongation following eEF2K inactivation.

This has important implications for the function of RPL24, showing that ribosomes in *Rpl24^Bst^* mutant cells can elongate efficiently despite the reduction in RPL24 expression. However, reduced RPL24 increases P-eEF2 which inhibits elongation, with an absolute requirement for eEF2K for this (**Figure 6C**). Importantly, the combination of the *Rpl24* and *eEF2K* mutants shows that when P-eEF2 is abolished elongation occurs at normal speed, despite reduced expression of RPL24.

eEF2K activity is regulated by several upstream signalling pathways, as well as factors such as calcium ion and oxygen levels (**Ballard et al., 2021**). Given that the *Rpl24^Bst^* mutation increases phosphorylation of P-eEF2, we assessed what effect the mutation had on multiple kinases upstream of eEF2K in *Apc^fl/fl^ Kras^G12D/+^* and *Apc^fl/fl^ Kras^G12D/+^ Rpl24^Bst/+^* small intestinal tissue and organoids. mTORC1 suppresses eEF2K activity via p70 S6K. However, we saw no difference in the mTORC1-regulated phosphorylation of RPS6 (S240/4) and 4E-BP1 (T37/46) (**Figure 2—figure supplement 1**). Similarly, MAPK signalling also suppresses eEF2K, via the RSK pathway. There was no difference in signalling through this pathway in *Rpl24* mutant mice, as assayed via P-ERK1/2 (T202/Y204) (**Figure 6—figure supplement 2A**). AMPK directly phosphorylates and activates eEF2K in response to changes in intracellular ATP:AMP ratio. Using phosphorylation of the canonical AMPK target acetyl-CoA carboxylase (P-ACC S79) as a read-out of AMPK activity, *Rpl24* mutation had no effect on AMPK signalling (**Figure 6—figure supplement 2B**). Thus, we find no evidence for RPL24 expression levels affecting the activity of three kinases that in turn regulate eEF2K.

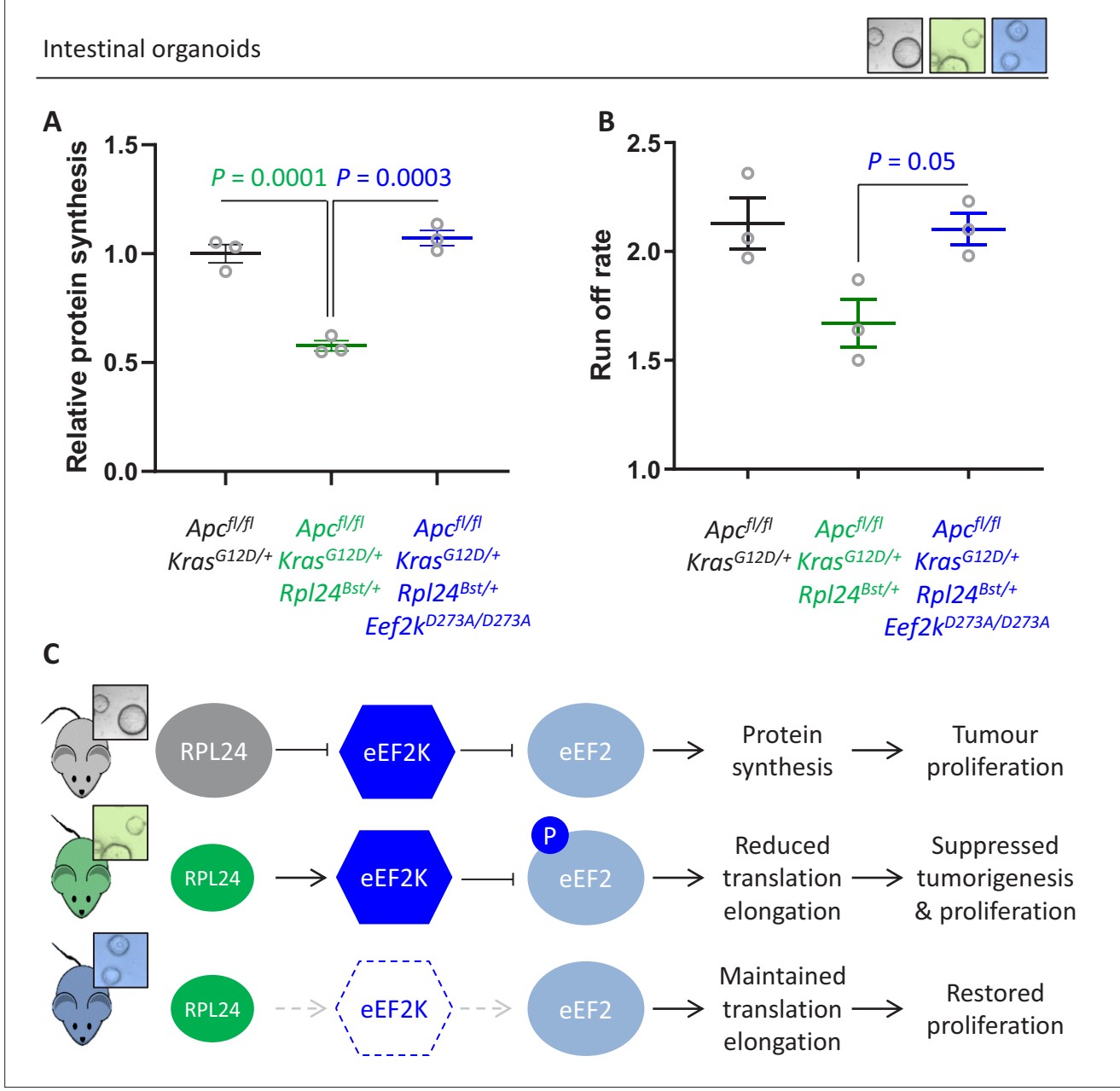

**Figure 6.** Genetic inactivation of *Eef2k* restores translation rates following *Rpl24^Bst* mutation. (**A**) ³⁵S-methionine incorporation to determine relative protein synthesis by in *Apc^fl/fl* *Kras^G12D/+* small intestinal organoids wild-type or mutant for *Rpl24* or with both *Rpl24* andConsistent with this, the *Eef2k* mutations. Data are represented ± standard error of the mean (standard error of the mean, SEM) with significance determined by one-way analysis of variance (ANOVA) with Tukey's multiple comparison. *N* = 3 per genotype, each representing an independent organoid line. (**B**) Ribosome run-off rate determined in
*Apc^fl/fl* *Kras^G12D/+* small intestinal organoids mutant or wild-type for *Rpl24* or with both *Rpl24* and *Eef2k* mutations. Data are the mean of three biologically independent organoid lines represented ± SEM with significance determined by Mann–Whitney *U* test. Raw data are available in *Figure 6—figure supplement 1A*. The run-off rate for *Apc^fl/fl* *Kras^G12D/+* control organoids is reproduced from *Figure 3E*. (**C**) Schematic representation of findings in *Apc*-deficient *Kras*-mutant mouse and organoid models. Top: RPL24 expression maintains translation and proliferation by suppressing the phosphorylation of eEF2 by limiting eEF2K activity. Middle: reduced expression of RPL24 activates eEF2K, increasing P-eEF2, reducing translation elongation and suppressing tumorigenesis and proliferation. Bottom: inactivation of eEF2K reverts the phenotype in *Rpl24^Bst* cells, due to the inability to phosphorylate and suppress eEF2. Elevated elongation rates correlate with increased proliferation following inactivation of eEF2K.

The online version of this article includes the following figure supplement(s) for figure 6:

**Figure supplement 1.** Inactivation of *Eef2k* restores translation elongation speed in *Rpl24^Bst* mutant mice.

*Figure 6 continued on next page*

*Figure 6 continued*

**Figure supplement 2.** *Rpl24^{Bst}* mutation has no effect on P-ERK or P-ACC.

**Figure supplement 2—source data 1.** Left: data from *Figure 6—figure supplement 2B*.

### The expression of *RPL24, EEF2K, and EEF2* is indicative of fast elongation in human CRC

Using pre-clinical mouse models, we have demonstrated that physiological RPL24 expression maintains low eEF2K-mediated phosphorylation of eEF2 (*Figure 7A*). Hypo-phosphorylated eEF2 then ensures rapid protein synthesis enabling tumour proliferation in vivo. We next sought to position this pre-clinical work in the context of clinical studies of the human disease. Using publicly available datasets for RNA expression in normal and cancerous colon tissues we observe increased *RPL24* and *EEF2* expression in conjunction with reduced *EEF2K* expression (*Figure 7B*). This mirrors the signalling pathways in our mouse models and ensures high expression of active eEF2. Elevated *EEF2* message and reduced *EEF2K* message levels in these clinical samples are consistent with conservation of this signalling in the clinic. Similar results are seen for the protein expression of RPL24, eEF2K, and eEF2 from colon adenocarcinoma samples (*Figure 7—figure supplement 1A*), and for the three mRNAs in rectal adenocarcinoma (*Figure 7—figure supplement 1B*). These expression analyses highlight the conservation of the proliferative tumour-associated signalling pathways characterised in our pre-clinical mouse models and patient samples.

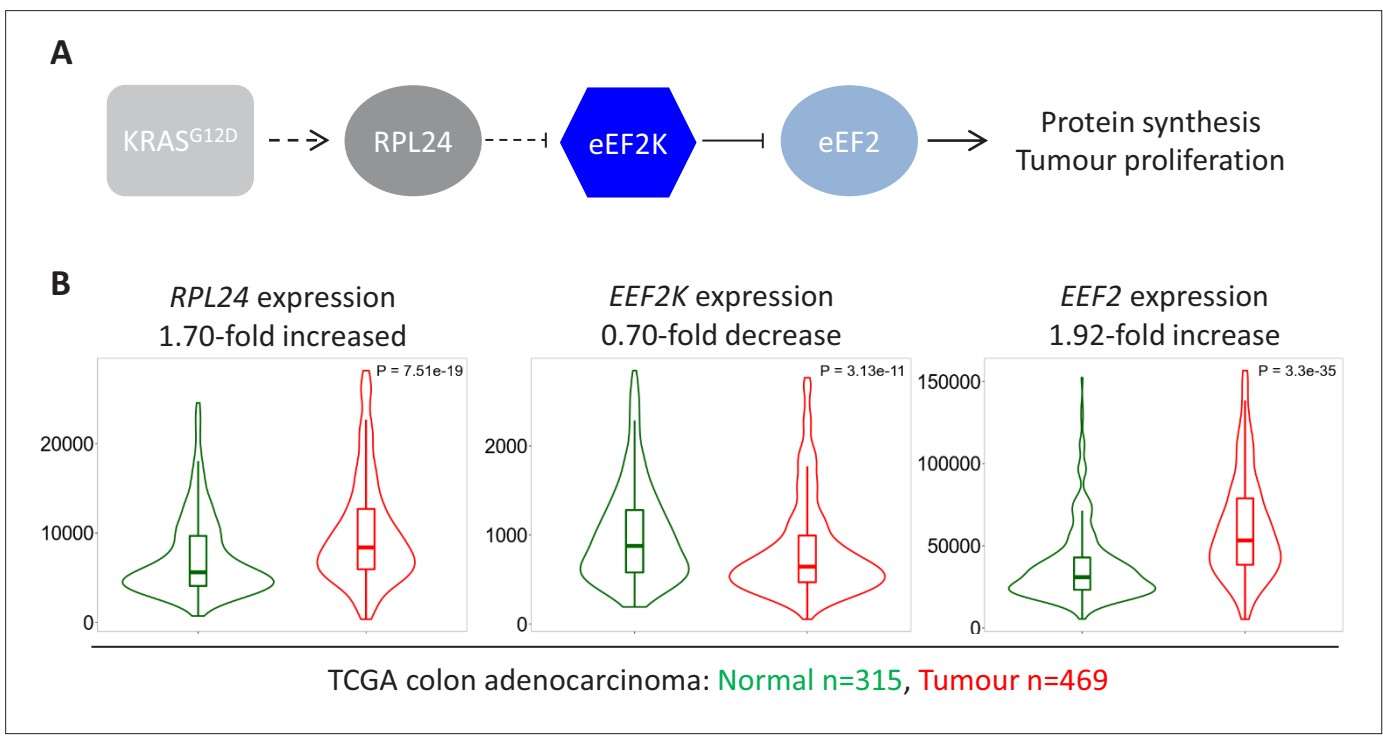

**Figure 7.** Expression of *RPL24, EEF2K,* and *EEF2* is consistent with increased eEF2 activity in colorectal cancer (CRC) tumours. (**A**) Schematic of the findings presented here from pre-clinical mouse models. KRAS activation requires RPL24 expression to maintain low eEF2 phosphorylation. This occurs via a double negative regulation of eEF2K, whereby RPL24 suppresses eEF2K, which suppresses eEF2. eEF2 activity correlates with protein synthesis and proliferation rates. Dashed lines indicate indirect or undefined regulatory pathways. (**B**) RNA expression levels of *RPL24, EEF2K,* and *EEF2* between normal colon and colon adenocarcinoma samples using data extracted from The Cancer Genome Atlas by TNMplot. Relative expression changes are annotated, as well as p values for each transcript.

The online version of this article includes the following figure supplement(s) for figure 7:

**Figure supplement 1.** Expression of *RPL24, EEF2K,* and *EEF2* suggest increased eEF2 activity in colorectal cancer (CRC) tumours.

## Discussion

The original characterisation of the *Rpl24^Bst* mutation identified a defect in ribosome biogenesis affecting the synthesis of 60S subunits (*Oliver et al., 2004*). The evidence to support this was limited to analyses of fasted/refed mouse livers for nascent rRNAs and by polysome profiles purporting to show reductions in 28S rRNA precursors and mature 60S subunits, respectively. In contrast, RNAi depletion of RPL24 in human cell lines had no effect on ribosome biogenesis, or on the relative abundance of 40S and 60S subunits (*Barkić et al., 2009*; *Wilson-Edell et al., 2014*). Furthermore, two reports have identified RPL24 as an exclusively cytoplasmic protein, leading to the hypothesis that it is assembled into mature ribosomes in the cytoplasm after rRNA synthesis and processing has already occurred (*Barkić et al., 2009*; *Saveanu et al., 2003*). In agreement with this, in the hindbrain of E9.5 embryos, the *Rpl24^Bst* mutation had no effect on nucleolar architecture, indicating that it is not required for nucleolar function (*Herrlinger et al., 2019*). Our data agree with these later examples, as we fail to see any effect of the *Rpl24^Bst* mutation on 60S:40S ratio in wild-type or transformed mouse intestines. Of further importance to the field, we also demonstrate in our tumour model that the translation defect in *Rpl24^Bst* mutant mice is restored to normal levels following inactivation of eEF2K in *Rpl24^Bst* mutant mice, showing that the defect is dependent on eEF2K. From this, we conclude that although the ribosomes produced in *Rpl24^Bst/+* mice do allow protein synthesis to proceed at near physiological levels, this process is restricted via signalling through eEF2K/P-eEF2.

RPL24 depletion suppresses tumorigenesis in *Apc*-deficient *Kras*-mutant (APC KRAS) mouse models of CRC, but not in *Apc*-deficient *Kras* wild-type (APC) models. In line with this, protein synthesis is reduced following depletion of RPL24 in our APC KRAS models, but not in APC models, despite induction of P-eEF2 in both cases. We previously showed that suppression of translation elongation via mTORC1/eEF2K/P-eEF2, using rapamycin, in the same APC models dramatically suppressed proliferation and extended survival (*Faller et al., 2015*). Thus, while RPL24 depletion or rapamycin treatment of *Apc*-deficient intestines each induce P-eEF2, only rapamycin treatment suppresses protein synthesis. The effect on protein synthesis appears to be the differential driving this divergence, with proliferation only impaired when protein synthesis is reduced. Crucially, RPL24 deficiency does not suppress mTORC1 activity since phosphorylation of the mTORC1 effectors 4E-BP1 and RPS6 (at S240/S244) is not reduced in *Apc*-deficient *Rpl24^Bst/+* intestines, providing an explanation of how proliferation and protein synthesis are maintained in *Apc*-deficient *Rpl24^Bst* mutant animals. We also provide evidence that *Rpl24* expression is lower than that of other ribosomal protein mRNAs following *Kras* mutation, which could explain why diminished RPL24 expression specifically suppresses proliferation in *Kras*-mutant models.

In view of the mechanisms we have described here, it is interesting to reflect on previous work with the *Rpl24^Bst* mutant mouse. The *Rpl24^Bst* mutation suppresses tumorigenesis in three mouse models of different types of blood cancer, and in a model of bladder cancer (*Signer et al., 2014*; *Barna et al., 2008*; *Hsieh et al., 2010*; *Jana et al., 2021*). In each case, the *Rpl24^Bst* mutation was found to decrease translation, although how the translation was suppressed was not fully determined. Barna et al. demonstrated a reduction in cap-dependent translation using a luciferase reporter in *Rpl24^Bst* MEFs (*Barna et al., 2008*). Furthermore, *Rpl24^Bst* mutation dramatically suppressed translation following MYC activation, consistent with *Rpl24^Bst* slowing tumorigenesis via reduced translation. We have shown that in mouse models of CRC the molecular mechanism by which normal levels of RPL24 maintain translation is via eEF2K and P-eEF2, therefore implicating this pathway in these previously studied blood cancer models.

The *Rpl24^Bst* mutation has also been used to analyse brain development from neural progenitor cells and explored as a model for retinal degenerative disease (*Riazifar et al., 2015*; *Herrlinger et al., 2019*). These studies found a revertant phenotype in neural progenitor cells overexpressing LIN28A and a defect in subretinal angiogenesis in *Rpl24^Bst/+* mice. The role of translation elongation should now be analysed with respect to these phenotypes. eEF2K, and the many upstream pathways it integrates, are potential targets for intervention in these in vivo models. It will also be of interest to determine how the inactivation of eEF2K affects the whole-body phenotypes of the *Rpl24^Bst* mouse, such as the coat pigmentation and tail defects.

eEF2K has a pleiotropic effect in tumorigenesis, acting akin to a tumour suppressor or promoter dependent on the context (*Knight et al., 2020b*). In some cancers, eEF2K promotes tumorigenesis. For example, under nutrient deprivation eEF2K acts as a pro-survival factor in transformed fibroblasts

and tumour cell lines by suppressing protein synthesis to ensure survival (*Leprivier et al., 2013*). Here, we show that eEF2K inactivation does not modify intestinal tumorigenesis. However, eEF2K is required for the suppression of tumorigenesis and protein synthesis following mTORC1 inhibition in APC cells (*Faller et al., 2015*) or *Rpl24*<sup>Bst</sup> mutation in APC KRAS cells (this work). Therefore, although eEF2K does not directly drive tumorigenesis, low eEF2K activity ensures there is no blockade of translation or proliferation in tumour cells. Furthermore, eEF2K expression is required for drug or signalling responses that suppress tumorigenesis, giving it tumour suppressive activity. In accordance, CRC patients with low eEF2K protein expression suffer a significantly worse prognosis (*Ng et al., 2019*). Furthermore, we present mRNA and protein expression data showing that eEF2K is reduced in clinical CRC samples compared to normal tissue. This agrees with a model where low eEF2K allows rapid translation elongation to promote proliferation.

Using the same clinical datasets we demonstrate that both RPL24 and eEF2 are elevated in CRC, consistent with their roles in promoting translation and proliferation. This presents the possibility of directly targeting either RPL24 or eEF2 for anti-cancer benefit. In agreement with this, RNAi against RPL24 in human breast cancer cell lines dramatically reduced proliferation (*Wilson-Edell et al., 2014*). Similarly, inhibiting eEF2, using a compound reported to slow its exit from the ribosome and thus the rate of protein synthesis, reduces cell line and patient derived organoid growth (*Stickel et al., 2015*; *Keysar et al., 2020*). These reports agree with the data presented here suggesting that targeting of RPL24 or eEF2 would be beneficial in CRC. Identifying the mechanistic link between RPL24 and eEF2K is part of our ongoing work, but here we have ruled out mTORC1, MAPK, and AMPK signalling as contributing factors.

This work uncovers an unexpected role for the ribosomal protein RPL24 in the regulation of translation elongation, acting via eEF2K/P-eEF2. We demonstrate that depletion of RPL24 suppresses tumorigenesis in a pre-clinical mouse model of a CRC. We provide genetic evidence supported by molecular assays of translation elongation to demonstrate that RPL24 depletion activates eEF2K to elicit tumour suppression in our models. We also speculate as to the role of eEF2K in the previously published blood cancer models where RPL24 depletion was beneficial. This work provides additional evidence for the anti-tumorigenic role of eEF2K in CRC, highlighting the potential for targeting translation elongation for this disease.

# Materials and methods

## Key resources table

| Reagent type (species) or resource | Designation | Source or reference | Identifiers | Additional information |
|---|---|---|---|---|
| Genetic reagent (*Mus musculus*) | Tg(Vil1-cre/ERT2)23Syr | *el Marjou et al., 2004* | RRID:MGI:3053826 | |
| Genetic reagent (*Mus musculus*) | Apc<sup>tm1Tno</sup> | *Shibata et al., 1997* | RRID:MGI:1857966 | |
| Genetic reagent (*Mus musculus*) | Kras<sup>tm4Tyj</sup> | *Jackson et al., 2001* | RRID:MGI:2429948 | |
| Genetic reagent (*Mus musculus*) | Rpl24<sup>Bst</sup> | *Oliver et al., 2004* | RRID:MGI:1856685 | |
| Genetic reagent (*Mus musculus*) | Eef2k<sup>D273A</sup> | *Gildish et al., 2012* | | |
| Genetic reagent (*Mus musculus*) | Apc<sup>Min</sup> | *Moser et al., 1990* | RRID:MGI:1856318 | |
| Cell line (*Mus musculus*) | Villin<sup>CreER</sup> *Apc*<sup>fl/fl</sup> *Kras*<sup>G12D/+</sup> small intestinal organoids | This study | | |
| Cell line (*Mus musculus*) | Villin<sup>CreER</sup> *Apc*<sup>fl/fl</sup> small intestinal organoids | This study | | |
| Cell line (*Mus musculus*) | Villin<sup>CreER</sup> *Apc*<sup>fl/fl</sup> *Kras*<sup>G12D/+</sup> *Rpl24*<sup>Bst/+</sup> small intestinal organoids | This study | | |
| Cell line (*Mus musculus*) | Villin<sup>CreER</sup> *Apc*<sup>fl/fl</sup> *Rpl24*<sup>Bst/+</sup> small intestinal organoids | This study | | |

*Continued on next page*

*Continued*

| Reagent type (species) or resource | Designation | Source or reference | Identifiers | Additional information |
|---|---|---|---|---|
| Antibody | BrdU (mouse monoclonal) | BD Biosciences #347,580 | RRID:AB_400326 | IHC: (1:250) |
| Antibody | P-eEF2 T56 (rabbit polyclonal) | Cell Signaling Technology #2,331 | RRID:AB_10015204 | WB: (1:2000) IHC: (1:100) |
| Antibody | eEF2 (rabbit polyclonal) | Cell Signaling Technology #2,332 | RRID:AB_10693546 | WB: (1:2000) |
| Antibody | P-4E-BP1 (rabbit monoclonal) | Cell Signaling Technology #2,855 | RRID:AB_560835 | IHC: (1:250) |
| Antibody | P-RPS6 S240/4 (rabbit monoclonal) | Cell Signaling Technology #5,364 | RRID:AB_10694233 | IHC: (1:100) |
| Antibody | P-eIF2$\alpha$ S51 (rabbit monoclonal) | Cell Signaling Technology #3,398 | RRID:AB_2096481 | IHC: (1:50) |
| Antibody | P-ERK T202/Y204 (rabbit polyclonal) | Cell Signaling Technology #9,101 | RRID:AB_331646 | IHC: (1:400) |
| Antibody | Lysozyme (rabbit polyclonal) | Dako A0099 | RRID:AB_2341230 | IHC: (1:300) |
| Antibody | RPS6 (mouse monoclonal) | Cell Signaling Technology #2,317 | RRID:AB_2238583 | WB: (1:2000) |
| Antibody | RPL10 (rabbit polyclonal) | Novus NBP1-84037 | RRID:AB_11007661 | WB: (1:2000) |
| Antibody | RPL22 (rabbit polyclonal) | Abcam ab111073 | RRID:AB_10863642 | WB: (1:2000) |
| Antibody | Acetyl-CoA carboxylase (rabbit monoclonal) | Cell Signaling Technology #3,676 | RRID:AB_2219397 | WB: (1:1000) |
| Antibody | P-acetyl-CoA carboxylase S79 (rabbit monoclonal) | Cell Signaling Technology #3,661 | RRID:AB_330337 | WB: (1:1000) |
| Antibody | β-Actin (mouse monoclonal) | Sigma-Aldrich #A2228 | RRID:AB_476697 | WB: (1:10,000) |
| Antibody | β-Tubulin (mouse monoclonal) | Cell Signaling Technology #2,128 | RRID:AB_823664 | WB: (1:4000) |
| Antibody | Goat Anti-Mouse Immunoglobulins/HRP (goat polyclonal) | Dako #P0447 | RRID:AB_2617137 | WB: (1:2000) |
| Antibody | Goat Anti-Rabbit Immunoglobulins/HRP (goat polyclonal) | Dako #P0448 | RRID:AB_2617138 | WB: (1:2000) |
| Sequence-based reagent | *Olfm4* RNAScope | ACD #311,838 | RNAScope | |
| Peptide, recombinant protein | Recombinant Murine Noggin | Peprotech #250-38 | | 100 ng/ml |

*Continued on next page*

*Continued*

| Reagent type (species) or resource | Designation | Source or reference | Identifiers | Additional information |
|---|---|---|---|---|
| Peptide, recombinant protein | Animal-Free Recombinant Human EGF | Peprotech #AF-100-15 | | 50 ng/ml |
| Peptide, recombinant protein | Recombinant Mouse R-Spondin 1 Protein | R&D Systems #3474-RS | | 500 ng/ml |
| Commercial assay or kit | Vectorstain Elite ABC-HRP | Vector Laboratories PK-6102 | RRIDs:AB_2336820 | |
| Commercial assay or kit | Vectorstain Elite ABC-HRP | Vector Laboratories PK-PK-6101 | RRIDs:AB_2336821 | |
| Commercial assay or kit | Cell Proliferation Labelling Reagent | Amersham Bioscience RPN201 | | |
| Commercial assay or kit | Cell-Titer Blue | Promega #G8080 | | |
| Chemical compound, drug | Rapamycin | LC Laboratories #R-5000 | | |
| Chemical compound, drug | Harringtonine | Santa Cruz sc-204771 | | |
| Chemical compound, drug | EasyTag EXPRESS 35S Protein Labeling Mix | Perkin Elmer #NEG772002MC | | |
| software, algorithm | Image J | *Rueden et al., 2017* | RRID:SCR_003070 | |
| software, algorithm | G*Power | *Faul et al., 2009* | RRID:SCR_013726 | |

## Materials availability

The mouse strains used will be made available on request. However, this may require a Materials Transfer Agreement and/or a payment if there is potential for commercial application. We ourselves are limited by the terms of Materials Transfer Agreements agreed to by the suppliers of the mouse strains.

## Mouse studies

Experiments with mice were performed under licence from the UK Home Office (licence numbers 60/4183 and 70/8646). All mice used were inbred C57BL/6J (Generation ≥8) and were housed in conventional cages with a 12 hr light/dark cycle and ad libitum access to diet and water. Mice were genotyped by Transnetyx in Cordova, Tennessee. Experiments were performed on mice between the ages of 6 and 12 weeks; both male and female mice were used. Sample sizes for all experiments were calculated using the G*Power software (*Faul et al., 2009*) and are shown in the figures or legends. Researchers were not blinded during experiments. The *Villin^{CreER}* allele (*el Marjou et al., 2004*) was used for intestinal recombination by intraperitoneal (IP) injection of tamoxifen in corn oil at a final in vivo concentration of 80 mg/kg. The *Apc* flox allele, *Apc^{Min}*, *Kras^{G12D}* lox-STOP-lox allele, *Eef2k^{D273A}* and *Rpl24^{Bst}* alleles were previously described (*Jackson et al., 2001*; *Shibata et al., 1997*; *Gildish et al., 2012*; *Oliver et al., 2004*; *Moser et al., 1990*). Tumour model experiments began with a single dose of tamoxifen after which mice were monitored until they showed signs of intestinal disease – paling feet from anaemia, weight loss and hunching behaviour. Tumours were scored macroscopically by counting and recording diameter after fixation of intestinal tissue. Tumour volumes were calculated from the tumour diameters assuming a spherical tumour shape. For short-term experiments, mice wild-type for *Kras* were induced on consecutive days (days 0 and 1) and sampled 4 days after the first induction (day 4). Mice bearing the *Kras^{G12D}* allele were induced once and sampled on day 3 post-induction. Where indicated, 250 µl of BrdU cell proliferation labelling reagent (Amersham Bioscience RPN201) was given by IP injection 2 hr prior to sampling. BrdU-positive cells were scored from crypt base to villus tip (a half crypt/villus axis) from the proximal small intestine or medial colon and

represented as the mean number of positive cells from at least 20 axes per mouse. For the regeneration experiments, mice were exposed to 10 Gy of γ-irradiation at a rate of 0.423 Gy/min from a caesium 137 source. They were then sampled 72 hr after irradiation and regenerative crypts scored from H&E stained sections as previously described (*Faller et al., 2015*). Batch correction was used by normalising each experiment to relevant wild-type controls.

## Histology and IHC

Tissue was fixed in formalin and embedded in paraffin. IHC staining was carried out as previously (*Faller et al., 2015*), using the following antibodies: BrdU (BD Biosciences #347580), P-eEF2 T56 (Cell Signaling Technology [CST] #2331), P-4E-BP1 T37/46 (CST #2855), P-RPS6 S240/4 (CST #5364), P-eIF2α S51 (CST #3398), P-ERK T202/Y204 (CST #9101), RPL24 (Sigma-Aldrich HPA051653), and Lysozyme (Dako A0099). The IHC protocol followed the Vector ABC kit (mouse #PK-6102, rabbit #PK-6101). RNAScope analysis was conducted according to the manufacturer's guidelines (ACD) using a probe to murine *Olfm4* (#311838). For all staining, a minimum of three biological replicates were stained and representative images used throughout. For BrdU scoring in short-term model experiments tissue was fixed in methanol:chloroform:acetic acid at a ratio 4:2:1 then transferred to formalin and embedded in paraffin.

## Intestinal organoid culture

Crypt cultures were isolated and then maintained as previously described (*Knight et al., 2020a*). In all crypt culture experiments, multiple biologically independent cultures were generated and analysed from different animals of the shown genotypes. DMEM/F12 medium (Life Technologies #12634-028) was supplemented with 5 mM HEPES (Life Technologies #15630-080), 100 U/ml penicillin/streptomycin (Life Technologies #1540-122), 2 mM L-glutamine (Life Technologies #25030-024), 1× N2, 1× B27 (Invitrogen #17502-048 and #12587-010), 100 ng/ml noggin (Peprotech #250-38), and 50 ng/ml EGF (Peprotech #AF-100-15). Wild-type cultures were also supplemented with 500 ng/ml R-spondin (R&D Systems #3474-RS). For ex vivo growth assays, cells were plated in technical triplicate in 96-well plates, and proliferation measured using Cell-Titer Blue (Promega #G8080) added to previously untreated cells each day for up to 4 days. Rapamycin (LC Laboratories R-5000) was dissolved in DMSO and administered for 24 hr, comparing to DMSO vehicle-treated cells.

## Western blotting

Samples were lysed (10 mM Tris [pH 7.5], 50 mM NaCl, 0.5% NP40, 0.5% SDS supplemented with protease inhibitor cocktail [Roche #11836153001] PhosSTOP [Roche #04906837001] and benzonase [Sigma-Aldrich #E1014]) on ice and then the protein content estimated by BCA assay (Thermo Fisher Scientific #23225). 20 μg of protein were denatured in loading dye containing SDS then resolved by 4–12% sodium dodecyl sulphate–polyacrylamide gel electrophoresis (Invitrogen #NP0336BOX). For western blots from gradients, equivalent volumes were loaded from each gradient fraction. Protein was transferred to nitrocellulose membranes, blocked with excess protein and immunoblotted overnight at 4°C using the following antibodies; P-eEF2 T56 (CST #2331), eEF2 (CST #2332), RPL24 (Sigma-Aldrich HPA051653), RPS6 (CST #2317), RPL10 (Novus NBP1-84037), RPL22 (Abcam ab111073), ACC (CST #3676), P-ACC S79 (CST #3661), β-tubulin (CST #2128), and β-actin (Sigma-Aldrich #A2228) as a sample control. One-hour incubation at room temperature with secondary antibodies (horseradish peroxidase [HRP]-conjugated anti-mouse secondary [Dako #P0447]; HRP-conjugated anti-rabbit secondary [Dako #P0448]) was followed by exposure to autoradiography films or a ChemiDoc MP imager (BioRad) with ECL reagent (Thermo Fisher Scientific #32106). Quantification was performed using Image J (*Rueden et al., 2017*).

## Sucrose density gradients

Cells were replenished with fresh medium for the 6 hr before harvesting. This medium was then spiked with 200 μg/ml cycloheximide (Sigma-Aldrich #C7695) 3 min prior to harvesting on ice. Crypt fractions from mice were isolated by extraction of the epithelium from 10 cm of proximal small intestine. Each data point plotted represents an individual animal. PBS-flushed linearly opened small intestines were incubated in RPMI 1640 medium (Thermo Fisher Scientific #21875059) supplemented with 10 mM EDTA and 200 μg/ml cycloheximide for 7 min at 37 °C with regular agitation to extract villi. Crypts

were isolated by transferring remaining tissue to ice-cold PBS containing 10 mM EDTA and 200 µg/ml cycloheximide for a further 7 min, again with agitation. The remaining tissue was discarded. Samples were lysed (300 mM NaCl, 15 mM MgCl2, 15 mM Tris pH 7.5, 100 µg/ml cycloheximide, 0.1% Triton X-100, 2 mM DTT, and 5 U/ml SUPERase.In RNase Inhibitor (Thermo Fisher Scientific #AM2696)) on ice and post-nuclear extracts placed on top of 10–50% wt/vol sucrose gradients containing the same buffer (apart from no Triton X-100, DTT, or SUPERase.In). These were then spun in an SW40Ti rotor at 255,000 rcf for 2 hr at 4°C under a vacuum. Samples were then separated through a live 254 nM optical density reader (ISCO). Polysome to subpolysome (P:S), heavy to light polysome (H:L), or 60S to 40S (60S:40S) ratios were calculated using the manually defined trapezoid method. For harringtonine run-off assays, cultures were prepared in duplicate for each genotype. One was pre-treated with 2 µg/ml harringtonine (Santa Cruz sc-204771) for 5 min (300 s) prior to cycloheximide addition. This was then processed as above. Run-off rates were calculated as previously described (*Knight et al., 2015*). Trichloroacetic acid protein precipitation from sucrose density gradients was performed as before (*Knight et al., 2013*).

### $^{35}$S-methionine incorporation assay

Organoids were replenished with medium 6 hr prior to analysis while in the optimal growth phase post-splitting. Technical triplicates were used from organoids from three different animals per experiments. $^{35}$S-methionine (Perkin Elmer #NEG772002MC) was used at 30 µCi/ml for 30 min. Samples were lysed using the same buffer described for Western blotting. Protein was precipitated in 12.5% (wt/vol) trichloroacetic acid onto glass microfiber paper (Whatmann #1827-024) by use of a vacuum manifold. Precipitates were washed with 70% ethanol and acetone. Scintillation was read on a Wallac MicroBeta TriLux 1450 scintillation counter using Ecoscint scintillation fluid (SLS Ltd #LS271) from these microfiber papers. In parallel the total protein content was determined by BCA assay using unprecipitated sample. Protein synthesis rate was expressed as the scintillation normalised to the total protein content (CPM/µg/ml protein), which was then changed to a relative value compared to relevant controls for each experiment.

### RNA sequencing

This was performed as previously described (*Knight et al., 2020a*), using three animals per genotype. The reads for each ribosomal protein mRNA were then averaged and the fold change compared to wild-type tissue calculated and plotted. Source data for this analysis are available linked to *Figure 4F*.

### Publicly available clinical data analysis

The TNMplot and UALCAN web portals were used to analysis publicly available dataset from The Cancer Genome Atlas and Clinical Proteomic Tumor Analysis Consortium. Details of these portals are available in these publications (*Bartha and Győrffy, 2021*; *Chandrashekar et al., 2017*).

### Statistical analyses

All statistical analyses are detailed in the relevant figure legends. In all cases, calculated p values less than or equal to 0.05 were considered significant. *N* numbers for each experiment are detailed within each figure, as individual points on graphs or within figure legends.

## Acknowledgements

The Sansom laboratory was funded by CRUK (A17196, A24388, and A21139), The European Research Council ColonCan (311301). This work was also funded by a Wellcome Trust Collaborative Award in Science (201487) to GM, CMS, TvdH, AEW, and OS. We are grateful to the Advanced Technologies and Core Services at the Beatson Institute (funded by CRUK C596/A17196 and A31287), particularly the Biological Services Unit, Histology Services and Transgenic Technology Laboratory. CGP is supported by funding from the National Health and Medical Research Council (Australia). We thank Daniel Murphy for kindly supplying antibodies for ACC and P-ACC. We thank Fiona Warrander for critical reading of the manuscript.

# Additional information

## Funding

| Funder | Grant reference number | Author |
|---|---|---|
| Cancer Research UK | A17196 | Owen J Sansom |
| Cancer Research UK | A24388 | Owen J Sansom |
| Cancer Research UK | A21139 | Owen J Sansom |
| H2020 European Research Council | 311301 | Owen J Sansom |
| Wellcome Trust | 201487 | Giovanna R Mallucci Tobias von der Haar Christopher Mark Smales Anne E Willis Owen J Sansom |
| National Health and Medical Research Council | | Christopher Proud |

The funders had no role in study design, data collection, and interpretation, or the decision to submit the work for publication.

## Author contributions

John RP Knight, Conceptualization, Formal analysis, Investigation, Methodology, Project administration, Writing - original draft, Writing - review and editing; Nikola Vlahov, David M Gay, Rachel A Ridgway, William James Faller, Investigation; Christopher Proud, Resources; Giovanna R Mallucci, Tobias von der Haar, Christopher Mark Smales, Conceptualization, Funding acquisition; Anne E Willis, Conceptualization, Funding acquisition, Writing - original draft, Writing - review and editing; Owen J Sansom, Conceptualization, Funding acquisition, Project administration, Supervision, Writing - original draft, Writing - review and editing

## Author ORCIDs

John RP Knight http://orcid.org/0000-0002-8771-5484
Tobias von der Haar http://orcid.org/0000-0002-6031-9254
Anne E Willis http://orcid.org/0000-0002-1470-8531
Owen J Sansom http://orcid.org/0000-0001-9540-3010

## Ethics

Experiments with mice were performed under a licence from the UK Home Office (licence numbers 60/4183 and 70/8646).

## Decision letter and Author response

Decision letter https://doi.org/10.7554/eLife.69729.sa1
Author response https://doi.org/10.7554/eLife.69729.sa2

# Additional files

## Supplementary files

• Transparent reporting form

## Data availability

Source data for Figure 1F, Figure 3 - figure supplement 1, Figure 3 - figure supplement 2, Figure 4F and Figure 6 - figure supplement 2 have been uploaded.

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
