## [Editor Report]

Briefly, Knight and colleagues investigate the role of the ribosome and translational control in colorectal tumours. A mutation of a protein of the large ribosomal subunit, RPL24, is used to suppress tumours driven by two mutations found commonly in cancer, in APC and KRAS. The authors identify a mechanistic output of the RPL24 BST mutation, eEF2 phosphorylation, which they demonstrate is a major effector in inhibiting tumour cell translation and proliferation. By targeting the eEF2 kinase eEF2K, they restore protein synthesis in RPL24 mutant cells. The conclusion is well supported by the experimental data presented, which implies that translation elongation can be a potential therapeutic target of KRAS mutated CRC. Importantly, Rpl24Bst in wildtype intestine does not affect epithelial cell proliferation and differentiation, suggesting that translation elongation can be used as tumour-specific target.

---

## [Decision Letter]

**Decision letter after peer review:**

Thank you for submitting your article "Rpl24Bst mutation suppresses colorectal cancer by promoting eEF2 phosphorylation via eEF2K" for consideration by *eLife*. Your article has been reviewed by 2 peer reviewers, and the evaluation has been overseen by a Reviewing Editor and David Ron as the Senior Editor. The following individual involved in review of your submission has agreed to reveal their identity: Vivian Li (Reviewer #1).

The Reviewing Editor has drafted this to help you prepare a revised submission.

The manuscript is well written with a strength being the design of experiments that combines genetically modified mouse models, organoid culture, biochemical assays, and transcriptomic analysis to address the role of RPL24 in CRC. The approach is thorough and comprehensive, and there is sufficient evidence provided to support the main conclusions. The only limitations of the work that the authors may wish to address are:

i) It remains unclear why and how the tumour suppressive role of Rpl24Bst is specific to KRAS G12D mutation only, considering that increased eEF2 phosphorylation is also observed in the KRAS wildtype model, and

ii) The mechanism by which RPL24-bst induces eEF2-phosphorylation. How direct is this and does it involve ribosome heterogeneity and/or translational stress?

Importantly this work enhances the concept of targeting translational control in tumours.

Essential revisions:

1. The authors provide convincing data showing that Rpl24Bst inhibits CRC proliferation and tumour growth by suppressing protein translation elongation via Eef2k-dependent eEF2 phosphorylation. They further show that the tumour suppressive role of Rpl24Bst is specific to Kras G12D mutation but not in Kras wildtype Apc-deficient animals. The Kras mutant specificity of Rpl24Bst model is very interesting, yet the underlying mechanism is largely unclear. Although we appreciate that the underlying full mechanism might be the scope of another story, it would be helpful if the authors were to elaborate on this point, taking into account the following considerations: In p.260-262, the authors suggest that Rpl24 expression might be sufficient for Apc-deleted models but not Kras mutation due to the limited upregulation of Rpl24 in Figure 4F. However, upregulation of eEF2 phosphorylation in the Apc-deleted models (Figure 4B) is apparently equivalent to the Kras mutant model (Figure S2A). Importantly, the authors show that abrogation of eEF2 phosphorylation via Eef2k inhibition has no effect on tumour growth in Apc-deleted only model (Figure S5B). Alternative explanation of Kras mutation specificity could be that eEF2 phosphorylation (and reduced translation elongation) is only required in Kras mutant tumours but not in Apc-deficient tumours. It will strengthen the hypothesis if the authors can evaluate whether translation elongation is affected in the Eef2k mutant and Apc-deficient tumours to see if translation elongation via Eef2k-p-eEF2-axis is important for Apc mutation.

2. The BrdU cell counting throughout the manuscript is normalised "per half crypt". BrdU positive staining often goes beyond the crypts towards villi in Apc mutant and Apc/Kras mutant models. It is unclear how half crypt is defined in hyperproliferative crypts (such as Figure 2C) between genotypes.

3. In Figure 4B, the BrdU staining is clearly reduced in Apcfl/flRpl24Bst/+ compared to Apcfl/fl intestine, yet quantitation shows no difference. Again, this may depend on how "BrdU^+^ cells per half crypt" is defined in different genotypes when there is massive crypt expansion in one but not the other.

4. In figure S1F, the authors show that mutated Rpl24 restricts irradiation-induced regeneration by quantifying regenerative crypts. How did the authors define "regenerative crypt" without staining? Also, how was the data normalised with all wildtype controls set at = 1?

5. The authors show that tumour numbers are not affected in Figure S2D and S5E. What about tumour size?

6. There are multiple figure citation errors in the manuscripts, particularly in result session including line 148-9/figure 1D, lines 243,244/figure4D, line 274/figureS2A, line 277/figure4D, line 279/figure4E and line 287/figureS4E. Please proofread.

7. The authors claim that there are 75% increase in P-eEF2 and 50% reduction in RPL24 expression in Figure 1F. Where's the quantitation?

8. Please provide p-value for Figure 4B (H-score of Rpl24 and p-eEF2) and Figure S6B.

9. Questions remain around the mechanism of action of RPL24-bst. Does RPL24-bst really not alter ribosome abundance? How are the polysome profiles normalised? Are cell equivalents loaded? Could the ratio of different poly fractions be maintained +/- RPL24-bst but total number of ribosomes be reduced?

10. What is the protein expression level of other RPLs when RPL24bst is expressed? This information for a few RPLs would either support the concept that ribosome number is maintained or would reveal a co-ordinated reduction in large ribosome subunit proteins.

11. Does reduced RPL24 expression lead to ribosome heterogeneity? If ribosome number is unaltered, but one subunit, RPL24, is reduced, presumably there is ribosome heterogeneity? Would this lead to translational stress?

12. What is the mechanistic link between RPL24 and eEF2? Any hints at how eEF2 phosphorylation is influenced by RPL24? How direct is the mechanism? Is translational stress involved?

---

## [Author Response]

Essential revisions:1. The authors provide convincing data showing that Rpl24Bst inhibits CRC proliferation and tumour growth by suppressing protein translation elongation via Eef2k-dependent eEF2 phosphorylation. They further show that the tumour suppressive role of Rpl24Bst is specific to Kras G12D mutation but not in Kras wildtype Apc-deficient animals. The Kras mutant specificity of Rpl24Bst model is very interesting, yet the underlying mechanism is largely unclear. Although we appreciate that the underlying full mechanism might be the scope of another story, it would be helpful if the authors were to elaborate on this point, taking into account the following considerations: In p.260-262, the authors suggest that Rpl24 expression might be sufficient for Apc-deleted models but not Kras mutation due to the limited upregulation of Rpl24 in Figure 4F. However, upregulation of eEF2 phosphorylation in the Apc-deleted models (Figure 4B) is apparently equivalent to the Kras mutant model (Figure S2A). Importantly, the authors show that abrogation of eEF2 phosphorylation via Eef2k inhibition has no effect on tumour growth in Apc-deleted only model (Figure S5B). Alternative explanation of Kras mutation specificity could be that eEF2 phosphorylation (and reduced translation elongation) is only required in Kras mutant tumours but not in Apc-deficient tumours. It will strengthen the hypothesis if the authors can evaluate whether translation elongation is affected in the Eef2k mutant and Apc-deficient tumours to see if translation elongation via Eef2k-p-eEF2-axis is important for Apc mutation.

We previously published on the importance of translation elongation in *Apc*-deficient mouse models of colorectal cancer, finding that rapamycin suppresses tumorigenesis in a mechanism entirely dependent upon eEF2K expression and signalling to P-eEF2 (Faller et al., Nature, 2015. PMID: 25383520). This is discussed in the introduction (lines 51-58), and again as we introduce the *Apc*-deficient models in the Results section (lines 257-260). Therefore, we can rule out the possibility that *Apc*-deficient models are less sensitive to suppression of elongation as an explanation for the lack of effect with the *Rpl24* mutant presented here, when compared to the *Apc*-deficient *Kras*-mutant models. As suggested by the reviewers we have quantified proliferation in Apc^fl/fl^, Apc^fl/fl^ Rpl24^Bst/+^, Apc^fl/fl^ Eef2k^D273A/D273A^ and Apc^fl/fl^ Rpl24^Bst/+^ Eef2k^D273A/D273A^ intestines, to test the effect of Eef2K deletion in this *Kras* wild-type model (Figure 5 —figure supplement 1D). We see no difference in proliferation in either the small intestine or colons of these genotypes, consistent with neither the Rpl24^Bst/+^ or Eef2k^D273A/D273A^ alleles having any effect on proliferation or tumorigenesis in this setting.

We sought to supplement the data on ribosomal protein expression across the different tumour models by quantifying protein levels by Western blot. This was carried out for RPL24, RPL22 and RPS6 expression in lysates from wild-type, Apc^fl/fl^ and Apc^fl/fl^ Kras^G12D/+^ organoids, shown in Author response image 1. This revealed a truncated form of RPL24 that was the dominant isoform in the Apc^fl/fl^ organoids, while the full-length version was dominant in wild-type and Apc^fl/fl^ Kras^G12D/+^ organoids. Investigating this truncation is part of our ongoing work, and we anticipate this will be linked to the differential effect on tumorigenesis in the *Kras* wild-type and mutant models. Truncated RPL24 has not been previously published but was the subject of a talk at the EMBL Protein Synthesis and Translation Control conference in September 2021. We feel that our data are too preliminary to include in this manuscript but are happy to present it here. The blots also show that RPL24 protein expression is not greatly increased in the Apc^fl/fl^ Kras^G12D/+^ organoids compared to Apc^fl/fl^ organoids, in line with the RNAseq data in Figure 4F. In contrast, RPL22 and RPS6 show increased expression in the *Kras* mutant organoids. This is consistent with the model proposed that RPL24 levels become limiting in the Apc^fl/(fl/+)^ Kras^G12D/+^ models, due to a relatively modest increase in expression in *Kras* mutant cells compared to the other ribosomal proteins.

**Author response image 1. sa2fig1:** A truncated form of RPL24 is expressed in Apc^fl/fl^ but not Apc^fl/fl^ Kras^G12D/+^ organoids. The significance of this in regulating tumorigenesis is currently under investigation. Fold changes in RP abundance from Apc^fl/fl^ to Apc^fl/fl^ Kras^G12D/+^ are shown relative to β-actin. For RPL24 expression both isoforms were quantified together. Two exposures are shown for RPL24, short on top and longer below.

2. The BrdU cell counting throughout the manuscript is normalised "per half crypt". BrdU positive staining often goes beyond the crypts towards villi in Apc mutant and Apc/Kras mutant models. It is unclear how half crypt is defined in hyperproliferative crypts (such as Figure 2C) between genotypes.

The reviewer is correct that this designation could be misinterpreted. BrdU positivity was quantified for entire crypt/villus axes in all cases, negating the effects of hyperproliferation. Therefore, labelling has been corrected to read ‘per half crypt/villus’ throughout the figures, legends, and manuscript. A sentence has also been added to the Materials and methods, Mouse studies section clarifying the scoring method used, indicating that proliferation was scored from crypt base to villus tip.

3. In Figure 4B, the BrdU staining is clearly reduced in Apcfl/flRpl24Bst/+ compared to Apcfl/fl intestine, yet quantitation shows no difference. Again, this may depend on how "BrdU^+^ cells per half crypt" is defined in different genotypes when there is massive crypt expansion in one but not the other.

The images shown in the original submission were difficult to interpret due to low intensity of the BrdU stain, specifically in the Apc^fl/fl^ intestine image. This has been replaced with a better image from which it can be seen that the extent of crypt/villus proliferation is similar to the Apc^fl/fl^ Rpl24^Bst/+^ example. It is important to remember that staining of the intravillus stroma, which is outside the intestinal epithelium and thus not scored, can alter the appearance of the BrdU positive zone in these images.

4. In figure S1F, the authors show that mutated Rpl24 restricts irradiation-induced regeneration by quantifying regenerative crypts. How did the authors define "regenerative crypt" without staining? Also, how was the data normalised with all wildtype controls set at = 1?

Regenerating crypts were scored from the H and E stained slides as previously described (Faller et al., Nature, 2015. PMID: 25383520). Author response image 2 shows an example of regenerating crypts on an H and E slide at high magnification. The images in Figure 1 —figure supplement 2C have also been enlarged to help the reader interpret the regenerating crypts. With regards to the normalisation of the data, this is due to differences between the four batches of experiments that were performed. In each case *Rpl24^Bst^* mutation suppressed regeneration but the baseline regeneration of the controls was varied. Thus, the number of regenerating crypts per circumference was normalised to the control mice for each batch. Author response image 2 shows the data before normalisation. Two sentences have been added to the Materials and methods clarifying this scoring technique.

**Author response image 2. sa2fig2:** Left, example HandE staining of regenerative small intestine, showing two regenerating crypts with red arrows. Right, regeneration data before batch normalisation. Colour coded circles show each of 4 batches in yellow (1v1), pink (1v1), blue (2v1) and red (1v1).

5. The authors show that tumour numbers are not affected in Figure S2D and S5E. What about tumour size?

Tumour volumes have been added adjacent to both figures. These follow the same trend as the tumour size, showing no differences within each tumour model. Text has been added to the Results section to reference these graphs and a sentence added to the methods describing the calculation of tumour volumes from recorded diameters.

6. There are multiple figure citation errors in the manuscripts, particularly in result session including line 148-9/figure 1D, lines 243,244/figure4D, line 274/figureS2A, line 277/figure4D, line 279/figure4E and line 287/figureS4E. Please proofread.

We apologise for this and have now proofread and corrected these citation errors.

7. The authors claim that there are 75% increase in P-eEF2 and 50% reduction in RPL24 expression in Figure 1F. Where's the quantitation?

This quantification has now been added to the Western blot in Figure 1F in the form of raw values for each lane and a summary figure next to the values. The figure legend has also been altered to describe the additional data.

8. Please provide p-value for Figure 4B (H-score of Rpl24 and p-eEF2) and Figure S6B.

*P* values have been added for Figure 4B – 0.05 for RPL24 and, after a larger batch of staining was performed, 0.048 for P-eEF2. *P* values have been added to Figure 6B, showing a significant increase in heavy polysome from Apc^fl/fl^ Kras^G12D/+^ to Apc^fl/fl^ Kras^G12D/+^ Rpl24^Bst/+^ intestines but not from Apc^fl/fl^ Kras^G12D/+^ Rpl24^Bst/+^to Apc^fl/fl^ Kras^G12D/+^ Rpl24^Bst/+^ Eef2k^D273A/D273A^ intestines. This lack of significance has been clarified in the main text.

9. Questions remain around the mechanism of action of RPL24-bst. Does RPL24-bst really not alter ribosome abundance? How are the polysome profiles normalised? Are cell equivalents loaded? Could the ratio of different poly fractions be maintained +/- RPL24-bst but total number of ribosomes be reduced?

The polysome profiles are normalised internally, with differing amounts of input material loaded. This is due to the need for rapid sample handling to retain polysome integrity. In our experience, slow sample processing, such as counting cells or weighing tissue, leads to either RNA degradation or ribosome run-off. As such, we use the fact that all the ribosomes will be represented in each trace, as either sub-polysomes or polysomes, and then present data as ratios normalised within each gradient. These ratios are then compared between genotypes. i.e., the 60S:40S ratio and sub-polysome:polysome ratios are calculated with the data from each gradient replicate and will be unchanged by the overall quantity of material analysed. It is therefore true that we cannot use the gradient traces to exclude the possibility that different genotypes have different overall quantities of ribosomes. However, we provide evidence supporting broadly similar levels of ribosomes below.

10. What is the protein expression level of other RPLs when RPL24bst is expressed? This information for a few RPLs would either support the concept that ribosome number is maintained or would reveal a co-ordinated reduction in large ribosome subunit proteins.

We have quantified ribosomal protein abundances for RPL24, RPL10, RPL22 and RPS6 in Apc^fl/fl^ Kras^G12D/+^ and Apc^fl/fl^ Kras^G12D/+^ Rpl24^Bst/+^ organoids, where equivalent total protein was loaded, and quantification normalised to β-actin protein expression. Here we saw increased expression of RPL10, reduced RPL22 and unchanged levels of RPS6 in the Apc^fl/fl^ Kras^G12D/+^ Rpl24^Bst/+^ organoids compared to Apc^fl/fl^ Kras^G12D/+^ controls (Figure 3 —figure supplement 1E). This supports the conclusion that ribosome subunit abundance is not suppressed by mutation of Rpl24, and there is actually increased expression of RPL10. In the text we avoid making this statement regarding ribosome abundances. Instead, we refer to relative levels of the ribosomal subunits as a readout for ribosome biogenesis defects, as was previously reported for the Rpl24 mutant mouse (Oliver et al., Development, 2004. PMID: 15289434).

11. Does reduced RPL24 expression lead to ribosome heterogeneity? If ribosome number is unaltered, but one subunit, RPL24, is reduced, presumably there is ribosome heterogeneity? Would this lead to translational stress?

To address the ribosome heterogeneity question and whether RPL24 may be absent from some ribosomes we analysed the abundance of RPL24 protein within sucrose density gradients – shown as Figure 3 —figure supplement 2. This shows that in Apc^fl/fl^ Kras^G12D/+^ Rpl24^Bst/+^ organoids, RPL24 protein was incorporated into 60S subunits, 80S ribosomes and polysomes. The distribution of RPL24 appears slightly altered towards polysomes, although this is not significant, with less RPL24 in the 60S subunit fraction. Data are presented as a percentage distribution of the total amount of each protein studied. It is therefore important to interpret this while considering the lower expression of RPL24 in the Apc^fl/fl^ Kras^G12D/+^ Rpl24^Bst/+^ organoids, meaning that a modest reduction could actually be much greater. In contrast to the changes in RPL24 distribution, RPL10 expression is consistent between the two genotypes. We discuss these additional data in the text and how it may relate to ribosome heterogeneity. We respond to the question regarding translational stress below.

12. What is the mechanistic link between RPL24 and eEF2? Any hints at how eEF2 phosphorylation is influenced by RPL24? How direct is the mechanism? Is translational stress involved?

Elucidating this mechanism is part of ongoing work which we believe falls beyond the scope of this study. However, we do provide data focused on the phosphorylation dependent regulation of eEF2K in Apc^fl/fl^ Kras^G12D/+^ and Apc^fl/fl^ Kras^G12D/+^ Rpl24^Bst/+^ organoids and tissue. eEF2K activity is regulated by mTORC1 and MEK signalling (inhibitory phosphorylation events) and AMPK (activation). We assayed the activity of these signalling pathways by measuring the phosphorylation of their canonical substrates, finding no differences between the two genotypes. This is shown in Figures 2 —figure supplement 2A (mTORC1) Figure 6 —figure supplement 2A (MEK) and Figure 6 —figure supplement 2B (AMPK). We discuss this data at length in the Results section and indicate that our ongoing work is searching for the mechanistic link.

The question regarding translation stress is an important one, and we have addressed by looking at the phosphorylation status of eIF2α across the models analysed in the manuscript. We saw that the Rpl24^Bst^ mutation had no effect on P-eIF2α in otherwise wild-type mice (Figure 1 —figure supplement 1A-B), the Apc^fl/fl^ Kras^G12D/+^model (Figure 2 —figure supplement 1A) or in the Apc^fl/fl^ model (Figure 4 —figure supplement 2C). From this we conclude that translation stress signalling via eIF2α is not altered by mutation of *Rpl24*. In turn, this highlights the specificity in the regulation of eEF2K as a means to suppress protein synthesis in these mutant mice. These results are presented and discussed throughout the Results section, with the point regarding specificity of eEF2K regulation also mentioned.